# Modelling the Impact of VAT Fiscality on Branch-Level Performance in the Construction Industry—Evidence from Romania

Cristina Elena Badiu (Cazacu) [1], Nicoleta Bărbuță-Mișu [2], Mioara Chirita [3], Ionica Soare [3], Monica Laura Zlati [2], Costinela Fortea [2] and Valentin Marian Antohi [1,2,*]

1   Department of Finance, Accounting and Economic Theory, Faculty of Economics and Business Administration, Transilvania University of Brasov, 500036 Brasov, Romania
2   Department of Business Administration, Faculty of Economics and Business Administration, Dunarea de Jos University of Galati, 800008 Galati, Romania
3   Departament of Economics, Faculty of Economics and Business Administration, Dunarea de Jos University of Galati, 800008 Galati, Romania
*   Correspondence: valentin.antohi@ugal.ro

**Abstract:** Fiscal policy stands as a crucial pillar of economic development through its economic financing function. The regulatory effects of fiscality have been shown to reduce the ripple effects of uncertainties on economic growth within the EU. Unlike the average European economy, the Romanian economy has exhibited particularities concerning economic growth (ranking highly in economic growth among European nations in absolute terms), partly due to a more assertive fiscal policy applied to a consumption-based economy affected by hyperinflation (especially in the last five calendar years). The research issue stems from the premise of the lack of predictability in Romanian fiscal policy and its implications for the business environment. Our aim is to develop an econometric model of the fiscal effects of VAT on the business performance of the construction sector in Romania for the period 2010–2021. The methods employed involve empirical analysis and the development of consolidated industry-level databases followed by econometric modeling using the multiple linear regression method. The results of the research demonstrate that financial independence and solvency promote excessive taxation in emerging markets and developing countries, such as Romania, being correlated with the macroeconomic evolution of the respective state. Additionally, the results indicate that tax pressure can constitute a barrier to the sustainable development of firms, with direct repercussions for consumers. Attractiveness to investors is also affected, remaining a priority for companies. The study's findings will enable the identification of the main impediments and opportunities brought about by VAT taxation on branch-level performance, proving useful for construction sector managers and fiscal policy makers in fostering sustainable industry development and establishing a sustainable fiscal regime to safeguard investors.

**Keywords:** fiscal policy; VAT; financial performance; sustainability; econometric model



## 1. Introduction

In accordance with European Council Directive No. 112/2006 (European Council 2006), a unified VAT system was established at the European level. The primary objective of the directive is to standardize the legislation across member states and create a consistent framework for implementing provisions related to the value-added tax system. The Council emphasizes that the implementation of a VAT system aims to minimize factors that could distort competition conditions at the European level. According to the European definition, a VAT system achieves optimal simplicity and neutrality when the tax is applied broadly, covering all stages of production, distribution, and service delivery (European Council 2006). This, in turn, fosters competitive neutrality by balancing tax burdens among member states, necessitating fiscal harmonization.

In the literature specific to the field, the harmonization process of the value-added tax within the European Union (Cnossen 2022), resulting in unified tax rates for goods and services, is highlighted. The EU ensures a nondiscriminatory character by establishing uniform characteristics for entities subject to taxation, regardless of the frequency of their taxable operations. These operations, termed VAT-taxable transactions, are categorized as such to facilitate commercial exchanges within the union, even if conducted by registered operators in a different state from the transaction location. To avoid jurisdictional conflicts, taxable transactions are considered to occur at the place of goods delivery or service rendering, falling within the seller's jurisdiction. Exceptions are explicitly outlined, particularly for the leasing of movable goods and certain services such as telecommunications, broadcasting, and television services. Taxable persons can deduct VAT paid for goods or services acquired for transactions subject to taxation in the state where the respective goods or services are purchased. Furthermore, VAT related to these transactions can be deducted from the VAT paid for taxed transactions. As a rule, VAT cannot be deducted for an economic activity exempt from VAT. Moreover, deductions can be limited or adjusted in certain situations. Additionally, the Council establishes the generating events and VAT liabilities as aspects that need to be harmonized among member states by introducing a common VAT system. Some exceptions (European Council 2006) are allowed to prevent the loss of tax revenues through the use of legislative capacities of states concerning the determination of the taxable value of goods and services regarding intra-community acquisitions.

Another aspect within the scope of VAT relates to the standard VAT rate, which, according to the Council, should have a minimum value of 15%, subject to review in each member state. According to research conducted by Kowal and Przekota (2021) in the last decade, VAT rates have increased in almost all European Union states, leading to the expansion of the "grey tax zone". Adopting the standard VAT rate should not lead to distortions in the single European market or create significant disparities in the field among member states. The effects of applying reduced VAT rates to locally provided services need to be assessed by the European Commission. Aspects related to new jobs, economic growth, and the functionality of the single market are the main considerations.

Another facet of implementing VAT at the European level concerns VAT exemptions, which target areas such as social services, healthcare, financial services, and insurance. A complementary aspect of this approach is exemptions with the right to deduct, focusing on exports of goods from member states to third countries, as well as deliveries of goods between member states. In cases where member states are significantly affected by fraud related to VAT, they may apply a general reversal of the VAT liability for a period, implying the transfer of VAT payment responsibility from the supplier to the customer.

The main European changes regarding VAT were aimed at directions such as simplifying VAT compliance, European convergence of VAT, easy access to goods and protective services for the population during the pandemic period such as a VAT exemption for COVID vaccine supplies, the application of reduced rates for diagnostic medical devices, and temporary exemptions for certain imports and deliveries during the pandemic. Additionally, the EU allowed some states to apply preferential VAT rates for certain products during the pandemic or advocated for the elimination of reduced rates applied to products with a negative impact on the environment. Digital development was considered in VAT regulation, setting some reduced rates for internet access deficit or for promoting the digital economy. Other strategic directions included the social economy and the development of the green economy.

In 2020, the expected VAT compliance gap in the EU was EUR 93 billion, a decrease of EUR 31 billion from the year before. Overall, Romania remained the country with the largest VAT compliance gap in 2020, trailed by Malta, Italy, and Greece. In 2020, the countries with the smallest VAT compliance gaps were still Finland and Estonia (European Commission 2023a). Notwithstanding conditions conducive to enhancing VAT compliance, the Romanian gap persisted at a significant level. In actual terms, Romania's GDP expanded by 34% between 2013 and 2021. By lowering the standard statutory rate by four percentage



points in January 2016 and an additional one percentage point in 2017, the Romanian government greatly decreased the VAT burden. The rate's significant decline has no discernible effect on the growth of the VAT compliance gap (European Commission 2023b).

The value-added tax (VAT) is the primary indirect tax (alongside excises) or consumption tax. VAT has supplanted almost all other consumption taxes. During a transaction, the tax is paid on the value of the good/service subject to that transaction. Regarding budget revenues, these represented 32.1% of Romania's GDP in 2021, showing an increase compared to 2020 due to inflation and the growth of the consumer economy (Ministry of Public Finance of Romania 2021). The value of budget revenues of RON 379 billion in 2021 includes VAT collections of over RON 79 billion, accounting for 20% of budget revenues (Ministry of Public Finance of Romania 2022).

We appreciate that this share is a significant one, VAT representing an important source contributing to the consolidation of the state budget. The main causes underlying VAT collection schemes are voluntary compliance and collection assurance measures adopted by the National Agency for Fiscal Administration (NAFA) through enforcement offices, with a precedent of unfavorable circumstances observed in 2020 based on the volume of deferred obligations for payment by economic agents estimated at around RON 4 billion (National Agency for Fiscal Administration of Romania 2020). These deferrals were caused by the economic blockade that occurred in 2020, which was overcome in 2021 due to economic reform measures and renewed economic growth in Romania.

Forecasts for the coming years express skepticism regarding voluntary compliance, attributed to the energy crisis and geopolitical conflicts. NAFA aims to enhance the value of accelerated VAT refunds through measures to reschedule payments and promote taxpayers' co-interest, fostering a resilient climate within regional tax administrations.

The specialized literature contains numerous research studies in the field of taxation, especially value-added tax (VAT), with a focus on reducing elements of fraud and error, increasing revenue collection for the state budget, and fiscal sustainability. It is evident that when used as a financial lever, taxation has a direct impact on economic performance; however, this aspect is not sufficiently developed in the specialized literature. The present research aims to clarify significant aspects of the functionality of the tax mechanism regarding VAT and its impact on performance in the construction sector in Romania, formulating the following research questions:

Q1—Does excessive taxation represent an obstacle to the growth of enterprise performance?

Q2—In the context of excessive taxation, is there a risk that the quality of services may be affected in favor of protecting investor interests?

Q3—Does excessive taxation pose a long-term threat to the sustainable development of businesses?

Q4—Can the optimal or suboptimal effect of fiscal policy regulations in Romania be determined through econometric modeling?

Thus, we aim to critically analyze the evolution of VAT fiscality in Romania and its impact on performance in the construction sector. This scientific endeavor is intended to be based on the following objectives:

O1. The critical study of specialized literature regarding the impact of fiscal policy on the sustainable economic development of enterprises.

O2. Building a representative sample of companies for the study of the fiscal impact on business performance in the construction sector in Romania.

O3. Modeling the fiscal impact of VAT on business development in the construction sector in Romania.

O4. Disseminating the results of the model.

This study continues with the presentation of the research results from specialized literature regarding the perception of VAT fiscal performance, the methodology's presentation and the conceptualization of the model equation, the dissemination of the model's results, and the formulation of relevant conclusions.

## 2. Literature Review

### 2.1. Approaches Regarding the Regulatory Role of VAT in the Economy

The value-added tax (VAT) imposed on goods and services in many countries worldwide plays a regulatory role in the economy by serving as a significant revenue generator for the government and funding public services. Moreover, VAT also acts as a tool to promote economic growth and regulate consumption patterns. The regulatory role of VAT in the economy is evident in its contribution to ensuring tax compliance and preventing tax evasion, promoting transparency and fairness within the fiscal system by establishing clear guidelines and procedures that businesses must adhere to. Additionally, it significantly contributes to the efficient collection and allocation of tax revenues, which are crucial for funding public services and economic development.

In the context of the challenges posed by rapid economic and social transformations, globalization, digitization, and other factors outlined by Trifan et al. (Trifan et al. 2023), governments globally are compelled to enhance the efficiency of fiscal systems. This involves implementing new tax collection techniques and utilizing fiscal resources effectively on a global scale (Cirman et al. 2022; Pop and Pelau 2017). Despite successful fiscal, administrative, and political reforms leading to gains in revenue mobilization, the distribution of these achievements remains unequal among states (Dom et al. 2022). Romania, for instance, has implemented a significant financial instrument, the Recovery and Resilience Plan, aiming for sustainable reform actions and investments until 2026 (European Commission 2021). The objectives include modernization, efficient service provision, effective tax collection, increased budget revenues, improved tax collection, and optimized budgetary expenses (National Agency of Fiscal Administration 2021).

According to the Annual Report on Taxation 2023 (European Commission 2023a), all member states have proposed a wide range of reforms for various types of taxes, both in terms of direct and indirect taxation, from VAT to environmental taxation, from personal income tax to corporate income tax. Many reforms have focused on environmental taxes and VAT related to the energy crisis, transitioning toward more sustainable environmental taxes and a more sustainable economy. The value-added tax (VAT) represents a significant source of revenue for member states in the European Union, serving as one of the most important funding sources for their government budgets. Within the European Union (EU), VAT, a tax implemented and applied in accordance with the EU VAT code, is mandatory in all its member states. This code ensures uniformity and fairness in the collection of value-added tax (VAT) throughout the entire European Union, while granting member states a certain degree of autonomy in setting their own VAT rates.

VAT systems operate by levying a tax on goods and services at each stage of the supply chain, ultimately transferring the burden of the tax to the final consumer. VAT rates can vary depending on the nature of the goods or services, and several member states provide lower rates for certain types of products. Currently, Luxembourg has the lowest VAT rate at 17%, while Hungary has the highest at 27% (Global VAT Compliance 2023). Each EU member state can decide on VAT and set its VAT rate for goods and services. EU legislation prohibits VAT rates of below 15%, and at least one of the two reduced rates must be above 5%. To ensure fairness and clarity in collecting value-added tax (VAT), the European Union (EU) has implemented strategies such as cross-border information exchange and increased transparency. The aim of these policies is to ensure fair competition for businesses and minimize the chances of VAT-related fraud. Holá et al. (2022) argued that VAT is an important tax within the EU that is significantly affected by tax evasion. Member states have implemented measures to combat fraud, such as VAT registration, to prevent this. The authors aimed to determine whether these actions improve VAT collection. The results of this research demonstrate that the value of the VAT gap is reduced by VAT listings, the use of credit cards, and low levels of corruption, according to the findings. However, the VAT difference is extended by a reverse charge mechanism for goods, final consumption, imports, and the implicit rate of consumption taxes.

Víghová (2022) analyzed tax evasion, which represents a predominant issue in the European Union, leading to substantial revenue losses. Value-added tax (VAT) is the most widespread type of tax evasion. According to the author, while completely eradicating tax evasion is impossible, implementing efficient legislation and rigorous tax supervision can help identify and combat it. The study relied on data obtained from the Financial Administration of the Slovak Republic, specifically investigating cases of value-added tax evasion and corporate income tax. The impact of tax laws on VAT fluctuations was analyzed by Chrysanthakopoulos and Tagkalakis (2023), who assessed corporate tax rates and personal income tax rates using a sample of 52 countries from 1985 to 2019. The authors revealed that implementing rules regarding budget balance and revenues amplifies fiscal policy fluctuations in line with economic cycles. Conversely, adopting rules regarding expenditures leads to a fiscal policy response that aligns with the prevailing economic conditions. The specific characteristics of tax legislation play an essential role in determining the extent to which fiscal policies are influenced by economic cycles.

### 2.2. Collection of VAT and Fiscal Sustainability Aspects

The efficient collection of value-added tax (VAT), coupled with a broader understanding of fiscal sustainability, plays a pivotal role in shaping the economic landscape of a country. This consumption tax, levied on goods and services at every stage of production and distribution, serves as a significant revenue source for governments. The revenue generated is crucial for funding diverse public services and infrastructure projects, thereby contributing to the reduction of budget deficits, fostering economic growth, and ensuring the stability of a country's public finances. The proper collection and management of VAT emerge as vital components in maintaining fiscal sustainability.

Anagnostou et al. (2023) conducted an analysis of the regional impact of macroeconomic and policy impulses in three Polish macro-regions, employing a multi-regional computable general equilibrium (CGE) model. The study's findings illustrate significant transregional differences regarding the transmission mechanisms of policies at the macro level, affecting production and regional components. The authors emphasize the equity–efficiency compromise of regional policy, demonstrating how spatially oriented spending measures can promote regional convergence or increase aggregate production. Poland's budgets, at a narrow level, and those of the broader European Union (EU) are predominantly funded through tax revenues.

At the European level, strategic directions mandate measures to reduce taxation and enhance tax collection. By contrasting the effects of various factors on tax collection in the EU and Poland, Pluskota (2022) aimed to determine the influencing factors. The conclusions indicate that although consumption and corruption have a different effect on tax collection in Poland compared to the EU, growth, trade, consumption, and corruption are all significant factors affecting collection in both the EU and Poland. Uryszek and Klonowska (2022) analyzed the degree of fiscal unsustainability in Poland and determined the fiscal deficit needed to stabilize the scale of public debt and achieve fiscal sustainability. The authors argued that by addressing fiscal deficits concerning value-added tax (VAT) and personal income tax (PIT), Poland can meet its current fiscal requirements and attain stability in terms of its fiscal situation. Discovering ways to eliminate the fiscal gap has the potential to significantly alter the current scenario. Poland could achieve fiscal sustainability by effectively collecting currently uncollected VAT and corporate income tax through the efforts of public bodies.

The principles of a cooperation-based approach regarding the payment of tax debts were identified by Siglé et al. (2022). They analyzed a sample of large businesses using data from surveys and tax audit results in the Netherlands. They highlighted the role of procedural justice and demonstrated that transparency positively influences compliance with corporate income tax, but not in the case of value-added tax (VAT). According to the authors, internal tax control also contributes to transparency and compliance. If there are no significant deficiencies in administrative capacity, Cnossen (2022) appreciated that the

efficiency of VAT, or actual revenues compared to potential revenues, should be uniform. Policy and compliance gaps contributing to VAT inefficiencies are considered residual variables. This suggests that calculating VAT inefficiency provides a more meaningful benchmark, especially in determining the policy gap. An analysis of Dutch VAT revenues, representative of VAT in other EU member states, was utilized by the author, who explored three approaches to improving VAT performance—reforming the common directive, transferring the VAT model to member states, and implementing a contemporary common VAT that can be approved by member states. Hindriks and Serse (2022) conducted an interesting analysis on the influence of this transitional and exogenous change in VAT, particularly its impact on the electricity market in Belgium. In an effort to assist low-income households, the Belgian government reduced the VAT rate from 21% to 6% in April 2014. A change in administration in September 2015 led to the increase and rebalancing of the VAT rate to 21%. Utilizing a selection of Belgian firms in the energy sector, the authors assessed the VAT transfer rate for residential energy costs using econometric techniques. The outcome demonstrates that the price of electricity changed by 100% as a result of both tax increases and decreases. The analysis indicates that VAT volatility has changed rapidly and predictably in relation to the fiscal–budgetary measures adopted.

### 2.3. VAT Efficiency and Its Impact on Performance

The importance of the value-added tax (VAT) in the tax system lies in its capacity to generate government revenue while minimizing the burden on individuals. VAT allows for the taxation of value added at each stage of production, ensuring that businesses bear a fair share of the tax burden. By taxing the value added at each production stage, VAT promotes efficiency by encouraging businesses to streamline their operations and reduce costs. This fosters productivity and innovation, leading to economic growth. Additionally, VAT's ability to generate government revenue while minimizing the burden on individuals enhances efficiency by ensuring a more equitable distribution of tax responsibility.

A group of authors, including Mascagni et al. (2023), have shown that the value-added tax (VAT) is intended to be a self-sufficient and efficient consumption tax. However, its administration can be costly and complicated. The authors assessed the possibilities of VAT in countries with lower incomes. They demonstrated that for proportional tax rates, VAT leads to disparities in tax obligations among businesses, and argued about the vulnerabilities affecting VAT effectiveness. Another interesting study was conducted by Benzarti and Tazhitdinova (2021), analyzing the impact of the value-added tax (VAT) on business flows using all legislative changes in the European VAT space from 1988 to 2016. According to the authors, even during significant adjustments to VAT, this indicates a minimum degree of VAT elasticity in business flows. Comparing these elasticities with those in the commercial literature, they are much lower. The results suggest that the probability of VAT distorting trade flows is minimal. The impact of changes in the consumption tax on prices and unit sales of durable goods in Germany, using microeconomic-level data, was analyzed by Buettner and Madzharova (2021). The authors demonstrated that changes in tax rates are fully transferred to prices, with a temporary increase in unit sales before implementation.

The phenomenon of tax evasion was analyzed by a group of authors, Fülöp et al. (2022), from both economic and social perspectives, as well as the role played by accounting—and accountants—in preventing and combating tax evasion in the case of entities in Romania. The authors developed a questionnaire to which 247 registered accounting professionals responded through the Body of Expert and Licensed Accountants of Romania (CECCAR). The data collected from the questionnaire were analyzed to prevent and combat tax evasion, recommending measures such as digitalization, simplification of tax procedures, and clarifying legislation (strengthening it). In a study conducted at the European level, Buettner and Tassi (2023) demonstrated the importance of reverse taxation, a technique used by EU member states to efficiently stop VAT refunds and withdrawals between businesses. The adoption of reverse taxation has been found to increase VAT refund claims in affected industries, supporting VAT fraud before the introduction of this mechanism, based on data from

VAT files in Germany. Prior to the implementation of reverse taxation, estimates indicate that losses due to VAT fraud in these sectors amounted to nearly 5% of VAT revenues.

In 2007, to combat value-added tax (VAT) fraud, the European Union proposed a strategy that was launched for a number of member states regarding the possibility of substantially changing the VAT system. One of the options discussed was the introduction of an optional reverse charge mechanism with an impact on businesses (European Commission 2007b).

In this regard, the European Commission initiated a public consultation on this matter to gather the opinions of businesses and assess the anticipated impact of such an option on the additional costs and/or benefits that the potential introduction of the optional reverse charge mechanism might cause (European Commission 2007a).

In an original study, the authors Stiller and Heinemann (Stiller and Heinemann 2023) demonstrated that zero-rating cross-border deliveries allowed tax evaders to avoid paying significant amounts of value-added tax (VAT) each year in the European Union. One of the most significant measures to counteract this scheme was represented by the optional reverse charge mechanism (RCM). Using trade data gap (TDG) asymmetries in international trade, the authors identified, for the period 2003–2019, a reduction in fraud in terms of VAT revenue of up to EUR 8.4 billion through the optional reverse charge mechanism (RCM). The authors explained the domino effect of introducing RCM in the European Union, highlighting a harmful shift of fraud from RCM member countries to non-RCM member countries, and proposed a unified approach to combating VAT fraud.

A study conducted by Afonso et al. (2021) analyzed the impact of structural tax reforms on the efficiency of public spending in 18 OECD economies from 2006 to 2017. According to the authors, the input efficiency scores average around 0.6%, and increases in tax rates negatively affect efficiency. Increases in tax bases contribute to improving efficiency, and during recessions, efficiency improves alongside increases in VAT tax bases. Thomas (2022) re-evaluated the widely accepted belief that value-added tax (VAT) is regressive, doing so using a fiscal microsimulation model created for 27 OECD nations. This analysis examines the first simultaneous approaches used in previous studies and highlights how economic types affect cross-country analysis when taxes (VAT) are compared to income. The author argued that calculating taxes (VAT) relative to expenditure—thus excluding economic influence—might provide a more accurate description of VAT's effect on performance distribution. According to the author, in most of the 27 OECD countries considered, VAT is approximately proportional and slightly progressive. An interesting conclusion of the research shows that long-term VAT systems (which rely on maintaining or even increasing VAT) might lead to minor regressivity.

The influence of credit and debit card usage on VAT compliance was analyzed by Alognon et al. (2021). The authors used annual national data for 26 European Union member states from 2000 to 2016. Utilizing spatial and temporal variations in card usage, combined with an examination of instrumental variables, the authors found that a 1% increase in cash payments reduces the VAT gap by 0.51 percentage points, while a 1% increase in cash withdrawals increases the gap by 0.6 percentage points. The authors proposed implementing more suitable measures for VAT compliance and considering potential vulnerabilities in tax administrations' capabilities to monitor transactions.

The role of non-cash payments in third-party reporting for value-added tax (VAT) compliance was studied by Madzharova (2020) in an interesting research study. The author believes that economies with well-developed financial institutions that ensure the traceability of digital payments serve as a deterrent to the suppression of sales, even in the absence of explicit policies using electronic payments for tax enforcement. Based on country-level data analyzed for the European Union in this study, it is shown that a 1% increase in the value of card payments relative to the gross domestic product (GDP) improves VAT performance by 0.05–0.09%. Additionally, according to Bohne et al. (2023), an estimated increase of 1pp. or 5.51% in the use of electronic money results in a 11.9% reduction in VAT compliance gaps. The authors analyzed the relationship between the

proliferation of non-cash or e-money payments and VAT compliance for European Union countries from 2001 to 2021, noting a negative correlation between electronic currency, its usage, and VAT evasion. The authors suggested that changes in transaction payment behavior, such as the adoption of non-cash payments, could generate significantly higher tax revenues by reducing non-compliance.

An interesting study conducted by Gazzani (2021) examined the use of flexible fiscal mechanisms such as VAT surtaxes, surtaxes on the import/manufacture of risky substances, and land taxes to implement new environmental tax reform. The aim is to reduce pollution and emissions without negatively impacting low-income households. The author presented evidence of the regressive economic effects of environmental tax reform in the European Union. They introduced a feedback mechanism to create a reimbursement system, such as reductions or cash transfers, to offset the regressive effect of the tax on consumer prices. The study's results target the reduction of inequalities caused by overtaxing low-income households through surtaxing existing taxes.

## 3. Methodology

We aim to conduct an impact analysis regarding the influence of fiscality on the development of the construction sector in Romania through economic and financial indicators analyzed dynamically from 2010 to 2021 (RISCO 2023). The selected indicators seek to identify the impact of fiscality by studying the fiscal effect, the effect of asset accumulation reflected by balance sheet indicators of assets, the effect of indebtedness on equity accumulations, and the financial effect by examining the results from the profit and loss account.

To study the fiscal effect, the following indicators were collected and analyzed: the amount of the collected VAT taxable base; the collected VAT value; the taxable base amount for deductible VAT; the value of deductible VAT reported at the end of the reporting period; the VAT payable balance, the cumulative VAT payment value; the VAT payment balance at the end of the reporting period; and the negative VAT balance at the end of the reporting period. For the analysis of the effect of asset accumulation reflected by balance sheet indicators, the following indicators were selected: the value of fixed assets recorded in the balance sheet; the value of receivables; and the total value of assets. Regarding the analysis of the effect of indebtedness on equity accumulations, the following indicators were selected: the value of other liabilities, including short-term tax liabilities and social security recorded in the balance sheet; the total value of short-term liabilities; the value of other liabilities, including long-term tax liabilities and social security recorded in the balance sheet; the total value of long-term liabilities; the total value of liabilities recorded in the balance sheet; and the value of equity. For the financial effect analysis, the following indicators were selected: the turnover value; the amount of expenses with other taxes, fees and payments; operating profit value; total income value; total expenditure value; gross profit value; and distinct analysis of reported gross profit and reported gross loss.

For this study, a sample of 100 firms operating in the construction sector in Romania was selected. The firms were included in the sample using the criterion of the number of employees (all the selected companies had more than 100 employees) from 2010 to 2021. Additional inclusion criteria were the total asset value of companies greater than RON 1,500,000 and turnover value greater than RON 1,600,000 throughout the analyzed period.

Exclusion criteria were as follows: companies not operating in the construction sector were excluded; companies with temporary interruptions in activity were excluded; and micro-enterprises were excluded.

The representativeness of the sample was analyzed using the Cochran formula for an accepted average error of 10% ($p$-value < 0.1) concerning the expected output at the construction sector level, where, according to Topfirme.ro (2023), there were 42,000 entities registered in the Commercial Registry as of 2022. The Z2 score of Student's test corresponding to the established statistical error threshold is 1.645, estimating a minimum sample of 70 firms to obtain statistically relevant sample values compared to the population of

construction sector firms in Romania. The estimated proportion of attributes present in the sampled population is 50% compared to the total population of firms.

$$n_0 = \frac{N \cdot S^2}{\frac{N \cdot e^2}{Z^2} + S^2} = \frac{42,000 \times 0.25}{\frac{42,000 \times 0.010}{2.706} + 0.25} = 67.541 \tag{1}$$

It follows that the selected sample of 100 firms meets the representativeness criteria and will generate outputs with a statistical representativeness of over 90% compared to the general sample of firms in the construction sector in Romania.

To determine the impact of VAT on the construction sector, we aim to create a model that, based on the historical evolution of economic and financial indicators of 100 firms in the construction sector, will depict, in a segregated manner, an image of the correlation between deductible VAT and collected VAT with the economic and financial indicators from 2010 to 2021. It should be noted that at the time of finalizing the work, the financial statements of firms for the year 2022 were not available.

Based on the analysis conducted during data centralization (Romanian Ministry of Public Finance 2023), we have selected the following economic and financial indicators for inclusion in the model (see Table 1):

**Table 1.** Model indicators.

| No. crt. | Indicator Name | Formula and Unit of Measure | Symbol |
|---|---|---|---|
| 1 | Fixed assets | Net value (RON) | AI |
| 2 | Claims | Net value (RON) | C |
| 3 | Current assets | Net value (RON) | AC |
| 4 | Total assets | Net value (RON) | AT |
| 5 | Other liabilities, including short-term tax and social security liabilities | Net value (RON) | ADTS |
| 6 | Total short-term liabilities | Net value (RON) | DTS |
| 7 | Other debts, including long-term tax and social security debts | Net value (RON) | ADTL |
| 8 | Total long-term liabilities | Net value (RON) | DTL |
| 9 | Total debts | Net value (RON) | DT |
| 10 | Equity | Net value (RON) | CP |
| 11 | Fiscal value | Net value (RON) | CA |
| 12 | Expenses with other taxes, fees and charges | Net value (RON) | TAX |
| 13 | Operating result | Net value (RON) | RE |
| 14 | Total income | Net value (RON) | VT |
| 15 | Total expenses | Net value (RON) | CT |
| 16 | Gross result | Net value (RON) | RB |
| 17 | Gross profit | Net value (RON) | PB |
| 18 | Gross loss | Net value (RON) | PrB |
| 19 | Number of employees | Persons | SAL |
| 20 | Return on equity (ROE) | Net result/Equity (%) | ROE |
| 21 | Return on assets (ROA) | Operating result/Total assets (%) | ROA |
| 22 | Equity solvency | Total assets/Total liabilities (%) | S |
| 23 | Financial independence | Equity/Total assets (%) | IF |
| 24 | VAT deductible | Net value (RON) | TVAD |
| 25 | VAT collected | Net value (RON) | TVAC |

Source: compiled by the authors.

The specialized literature presents a varied case study regarding VAT, focusing mainly on reducing fraud and errors (Bohne et al. 2023; Buettner and Tassi 2023; Cirman et al. 2022; Fülöp et al. 2022; Stiller and Heinemann 2023), increasing revenue collection for the state

budget (Kowal and Przekota 2021; Madzharova 2020), and ensuring fiscal sustainability (Uryszek and Klonowska 2022). It is clear that when used as a financial method, taxation has a direct impact on economic performance. However, this aspect is not sufficiently explored in the specialized literature.

The following research hypotheses are to be investigated during the modeling:

**H1.** *Financial independence and solvency indicators vary inversely proportionally to the evolution of VAT at the industry level, implying that in Romania, excessive fiscality represents a significant obstacle to the economic development of entities.*

**H2.** *The level of profitability expressed synthetically through rates of return on capital and assets varies inversely proportionally to the evolution of collected and deductible VAT, meaning that excessive fiscality is a major hindrance to the sustainable development of firms in Romania.*

**H3.** *Rates of return on equity are less sensitive to the tightening of the VAT regime than rates of return on assets, indicating that in Romania, the tightening of the tax regime leads to a transfer of savings at the expense of the quality offered to customers, while protecting investors' interests remains a priority.*

**H4.** *Analytical economic and financial indicators tend to follow the trend of direct correlation with VAT fiscal policy, meaning that in Romania, the level of regulation of VAT fiscal policy is optimal.*

To demonstrate the previously stated hypotheses, we have defined and constructed the following economic model:

$$
\begin{cases}
\begin{aligned}
VATD_i = & \propto_{1_i} * AI_i + \propto_{2_i} * C_i + \propto_{3_i} * AC_i + \propto_{4_i} * ADTS_i + \propto_{5_i} * DTS_i + \propto_{6_i} * ADTL_i + \propto_{7_i} * DTL_i \\
& + \propto_{8_i} * DT_i + \propto_{9_i} * CP_i + \propto_{10_i} * CA_i + \propto_{11_i} * TAX_i + \propto_{12_i} * RE_i + \propto_{13_i} * CT_i \\
& + \propto_{14_i} * PB_i + \propto_{15_i} * PrB_i + \propto_{16_i} * SAL_i + \propto_{17_i} * ROE_i + \propto_{18_i} * ROA_i + \propto_{19_i} * S_i \\
& + \propto_{20_i} * IF_i + \epsilon_i \\
VATC_i = & \propto_{1_i} * AI_i + \propto_{2_i} * C_i + \propto_{3_i} * AC_i + \propto_{4_i} * ADTS_i + \propto_{5_i} * DTS_i + \propto_{6_i} * ADTL_i + \propto_{7_i} * DTL_i \\
& + \propto_{8_i} * DT_i + \propto_{9_i} * CP_i + \propto_{10_i} * CA_i + \propto_{11_i} * TAX_i + \propto_{12_i} * RE_i + \propto_{13_i} * CT_i \\
& + \propto_{14_i} * PB_i + \propto_{15_i} * PrB_i + \propto_{16_i} * SAL_i + \propto_{17_i} * ROE_i + \propto_{18_i} * ROA_i + \propto_{19_i} * S_i \\
& + \propto_{20_i} * IF_i + \epsilon_i
\end{aligned}
\end{cases} \tag{2}
$$

where:

TVAC—the dynamics of the collected VAT value in the sector in year *i* (dependent variable);
TVAD—the dynamics of the deductible VAT value in the sector in year *i* (dependent variable);
*i*—the seasonal component of VAT variation;
$\propto_{x_i}$—the seasonal regression coefficients of the economic and financial regressors;
*x*—the number of the indicator for which the regression coefficient is calculated;
$\epsilon_i$—the seasonal residual variable.

After modeling, the general equation could be defined validly in seasonal terms, as follows:

It is observed that there are significant differences between the two structural models of VAT, according to Figure 1. The values of the unstandardized β coefficients of the two OLS regressions depicted in Figure 1 are presented in Figure 2.

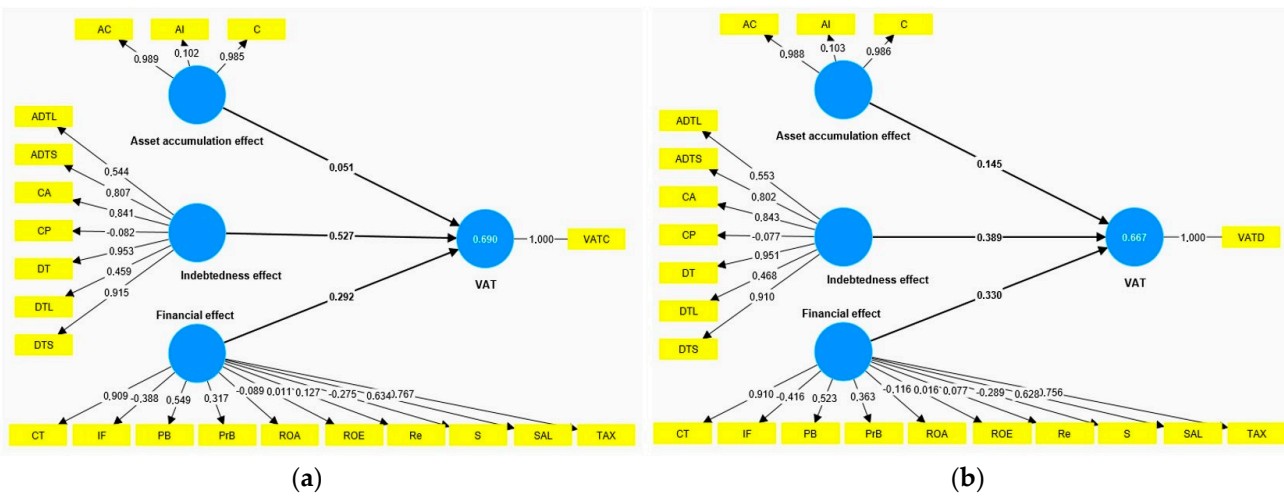

**Figure 1.** Structural model—(**a**) collected VAT (VATC) vs. (**b**) deductible VAT (VATD). Source: created by the authors using SmartPLS4 software V4.

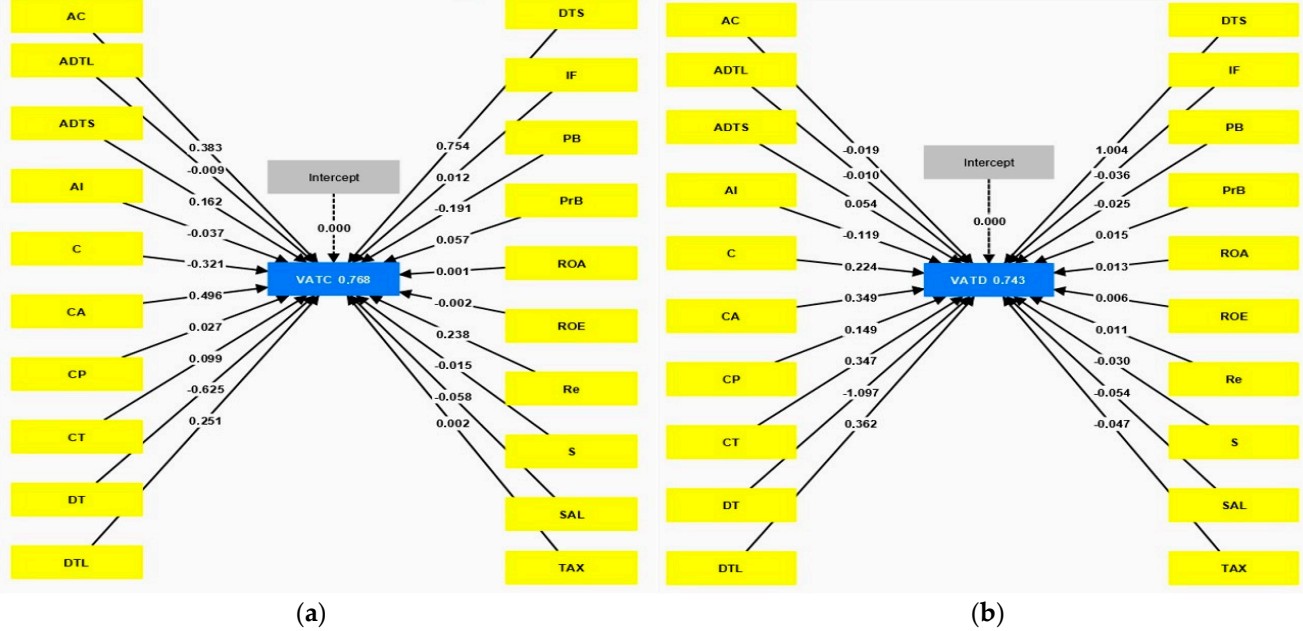

**Figure 2.** Unstandardized coefficients of OLS regressions—(**a**) collected VAT (VATC) vs. (**b**) deductible VAT (VATD). Source: created by the authors using SmartPLS4 software V4.

The analysis of the dynamic heterogeneous distribution of deductible VAT and collected VAT confirms the instability and unpredictability of the business environment in terms of voluntary compliance with VAT (Figure 3).

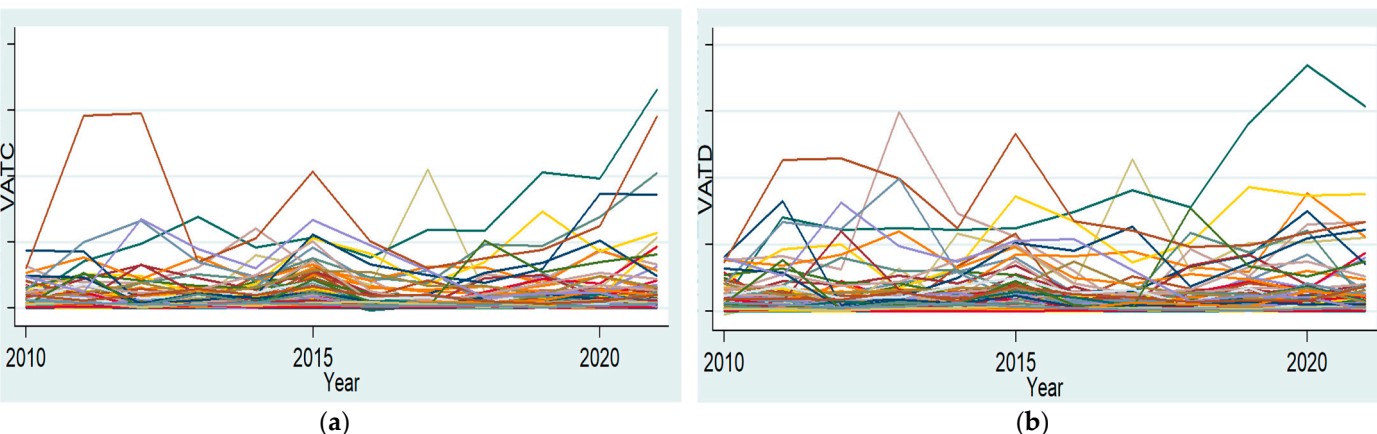

**Figure 3.** Seasonal trend variations—(**a**) collected VAT (VATC) vs. (**b**) deductible VAT (VATD). Source: created by the authors using STATA software V13.

Validation tests of the structural equation model have been conducted, indicating that the models can be accepted both in terms of absolute values and through the incremental fit and parsimonious fit perspectives of the estimated VATC and VATD models (Table 2).

**Table 2.** Model summary.

| Model Fit | | VATC | | VATD | |
|---|---|---|---|---|---|
| | | Saturated Model | Estimated Model | Saturated Model | Estimated Model |
| Absolute fit | SRMR | 0.0815 | 0.0815 | 0.0762 | 0.0762 |
| Incremental fit | NFI | 0.9375 | 0.9375 | 0.9350 | 0.9350 |
| Parsimonious fit | Chisq/df | 2.8367 | 2.8367 | 2.8323 | 2.8323 |

Source: compiled by the authors using *SmartPLS4* statistical software.

From Table 2, it can be observed that the standardized root mean square residual values, which determine the absolute mean of residual covariance (Hu and Bentler 1999), demonstrate a good fit of the model. Additionally, the value of the normal fit index (NFI > 0.9) indicates an acceptable fit of the model, and the chi-square/degrees of freedom test (<3) suggests a good parsimonious fit and acceptance of the models.

## 4. Results and Discussions

It can be observed from Table 3 that the representativeness level of the regressions varies between 60.5% and 75% for the VATC model and between 56.5% and 72.9% for the VATD model, respectively, thus demonstrating that the fiscal effect (VATC model) tends to exceed profitability functions estimated through the three effects—accumulation, indebtedness, and financial.

**Table 3.** The comparative summary of OLS regressions regarding the fiscal effect in relation to the performance of entities in the construction sector expressed through the cumulative effect of assets, leverage effect, and financial effect.

| ASSET ACCUMULATION EFFECT | | | | | | | | | | |
|---|---|---|---|---|---|---|---|---|---|---|
| VATC | Coef. | St.Err. | t-value | *p*-value | Sig | VATD Coef. | St.Err. | t-value | *p*-value | Sig |
| AI | 0.0010 | 0.0010 | 1.4700 | 0.1430 | | 0.0010 | 0.0000 | 1.6900 | 0.0920 | * |
| C | −0.0230 | 0.0030 | −8.0000 | 0.0000 | *** | −0.0070 | 0.0020 | −3.7500 | 0.0000 | *** |
| AC | 0.0420 | 0.0020 | 20.2700 | 0.0000 | *** | 0.0200 | 0.0010 | 15.3500 | 0.0000 | *** |
| Constant | 879,344.25 | 131,124.47 | 6.71 | 0 | *** | 648,235.79 | 83,073.313 | 7.8 | 0 | *** |
| Mean dependent var | | 3,471,659.8760 | SD dependent var | 6,206,870.3820 | | Mean dependent var | | 2,112,319.4160 | SD dependent var | 3,744,236.8190 |
| R-squared | | 0.6050 | Number of obs | 1200.0000 | | R-squared | | 0.5650 | Number of obs | 1200.0000 |
| F-test | | 611.5110 | Prob > F | 0.0000 | | F-test | | 517.1820 | Prob > F | 0.0000 |
| Akaike crit. (AIC) | | 39,835.5470 | Bayesian crit. (BIC) | 39,855.9070 | | Akaike crit. (AIC) | | 38,740.1300 | Bayesian crit. (BIC) | 38,760.4900 |

| INDEBTEDNESS EFFECT | | | | | | | | | | |
|---|---|---|---|---|---|---|---|---|---|---|
| VATC | Coef. | St.Err. | t-value | *p*-value | Sig | VATD Coef. | St.Err. | t-value | *p*-value | Sig |
| ADTS | 0.0280 | 0.0050 | 5.5000 | 0.0000 | *** | 0.0140 | 0.0030 | 4.3800 | 0.0000 | *** |
| DTS | 0.0300 | 0.0100 | 2.9300 | 0.0030 | *** | 0.0230 | 0.0070 | 3.5200 | 0.0000 | *** |
| ADTL | −0.0100 | 0.0110 | −0.9400 | 0.3480 | | −0.0030 | 0.0070 | −0.4400 | 0.6600 | |
| DTL | 0.0310 | 0.0110 | 2.8900 | 0.0040 | *** | 0.0250 | 0.0070 | 3.6800 | 0.0000 | *** |
| DT | −0.0240 | 0.0100 | −2.3200 | 0.0210 | ** | −0.0200 | 0.0070 | −3.0000 | 0.0030 | *** |
| CP | 0.0000 | 0.0010 | 0.2900 | 0.7750 | | 0.0010 | 0.0010 | 0.9700 | 0.3320 | |
| CA | 0.0260 | 0.0010 | 29.1600 | 0.0000 | *** | 0.0160 | 0.0010 | 28.2000 | 0.0000 | *** |
| Constant | −327,605.78 | 117,907.46 | −2.78 | 0.006 | *** | −192,978.37 | 74,320.309 | −2.6 | 0.01 | *** |
| Mean dependent var | | 3,471,659.8760 | SD dependent var | 6,206,870.3820 | | Mean dependent var | | 2,112,319.4160 | SD dependent var | 3,744,236.8190 |
| R-squared | | 0.7510 | Number of obs | 1200.0000 | | R-squared | | 0.7290 | Number of obs | 1200.0000 |
| F-test | | 514.7320 | Prob > F | 0.0000 | | F-test | | 457.1230 | Prob > F | 0.0000 |
| Akaike crit. (AIC) | | 39,288.8940 | Bayesian crit. (BIC) | 39,329.6150 | | Akaike crit. (AIC) | | 38,181.2560 | Bayesian crit. (BIC) | 38,221.9770 |

**Table 3.** *Cont.*

| | | | | | FINANCIAL EFFECT | | | | | |
|---|---|---|---|---|---|---|---|---|---|---|
| VATC | Coef. | St.Err. | t-value | *p*-value | Sig | VATD Coef. | St.Err. | t-value | *p*-value | Sig |
| TAX | 0.0400 | 0.1730 | 0.2300 | 0.8160 | | −0.1360 | 0.1060 | −1.2800 | 0.2000 | |
| Re | 0.0210 | 0.0430 | 0.5000 | 0.6190 | | −0.0250 | 0.0260 | −0.9400 | 0.3500 | |
| CT | 0.0320 | 0.0010 | 36.3200 | 0.0000 | *** | 0.0190 | 0.0010 | 35.3700 | 0.0000 | *** |
| PB | 0.0030 | 0.0410 | 0.0700 | 0.9470 | | 0.0400 | 0.0250 | 1.5900 | 0.1110 | |
| PrB | −0.0100 | 0.0420 | −0.2400 | 0.8100 | | −0.0290 | 0.0260 | −1.1000 | 0.2710 | |
| SAL | −1310.4830 | 396.6340 | −3.3000 | 0.0010 | *** | −581.7590 | 243.7590 | −2.3900 | 0.0170 | ** |
| ROE | −2102.3030 | 19,459.2190 | −0.1100 | 0.9140 | | 2413.8430 | 11,959.0270 | 0.2000 | 0.8400 | |
| ROA | 73,300.0500 | 797,372.0400 | 0.0900 | 0.9270 | | −14,336.9750 | 490,039.9200 | −0.0300 | 0.9770 | |
| S | 22,332.5460 | 43,253.2740 | 0.5200 | 0.6060 | | −14,734.1570 | 26,582.1100 | −0.5500 | 0.5790 | |
| IF | −1,168,691.0000 | 428,243.7400 | −2.7300 | 0.0060 | *** | −571,175.7800 | 263,185.2100 | −2.1700 | 0.0300 | ** |
| Constant | 264,127.74 | 227,912.79 | 1.16 | 0.247 | | 235,329.66 | 140,068.07 | 1.68 | 0.093 | * |
| Mean dependent var | | 3,471,659.8760 | | SD dependent var | 6,206,870.3820 | Mean dependent var | | 2,112,319.4160 | SD dependent var | 3,744,236.8190 |
| R-squared | | 0.7150 | | Number of obs | 1200.0000 | R-squared | | 0.7040 | Number of obs | 1200.0000 |
| F-test | | 298.6370 | | Prob > F | 0.0000 | F-test | | 283.3860 | Prob > F | 0.0000 |
| Akaike crit. (AIC) | | 39,457.9470 | | Bayesian crit. (BIC) | 39,513.9380 | Akaike crit. (AIC) | | 38,289.5440 | Bayesian crit. (BIC) | 38,345.5350 |

*** $p < 0.01$, ** $p < 0.05$, * $p < 0.1$. Source: realized by the authors with the help of STATA software V13.

Based on the data from Table 3 regarding the accumulation effect, the value of the F function for the tax effect is strictly higher than the F-function value for profitability (611 > 517), with the correlation level of independent variable coefficients being positive (direct correlation), except for receivables. For both functions, the Sig coefficients allow for the rejection of the null hypothesis, with recorded values of error representativeness below the chosen representativeness threshold of 0.1, except for fixed assets in the VATC version (tax effect), where a $p$-value > 0.1 is recorded. Regarding the leverage effect, it is observed that the level of the F function is higher in the case of VATC (tax effect) than in the case of VATD (profitability, 514 > 457). The significance level of the correlation between independent variables and the two dependent variables is high, except for other debts, including long-term tax and social security debts (ADTL) and equity (CP). The last effect analyzed is the financial effect, observing that, unlike the accumulation and leverage effects, it generates F-function values that are significantly closer (VATC > VATD, 298 > 283), and the number of variables with a high level of significance decreases compared to previous models. For correlation analysis and the estimation of time-fixed effects, correlation tests were conducted using STATA software V13, and the results are presented in Table 4.

**Table 4.** Time-fixed effects of OLS.

| VATC | Coef. | St.Err. | t-Value | $p$-Value | Sig | VATD | Coef. | St.Err. | t-Value | $p$-Value | Sig |
|---|---|---|---|---|---|---|---|---|---|---|---|
| AI | −0.002 | 0.002 | −0.93 | 0.351 | | AI | −0.001 | 0.001 | −1.12 | 0.262 | |
| C | −0.009 | 0.004 | −2.28 | 0.023 | ** | C | 0.002 | 0.002 | 0.67 | 0.504 | |
| AC | 0.008 | 0.004 | 2.2 | 0.028 | ** | AC | 0.007 | 0.002 | 2.92 | 0.004 | *** |
| ADTS | 0.032 | 0.006 | 5.01 | 0 | *** | ADTS | −0.005 | 0.004 | −1.27 | 0.206 | |
| DTS | 0.03 | 0.011 | 2.72 | 0.007 | *** | DTS | 0.017 | 0.007 | 2.49 | 0.013 | ** |
| ADTL | 0.029 | 0.013 | 2.26 | 0.024 | ** | ADTL | 0.007 | 0.008 | 0.86 | 0.389 | |
| DTL | 0.023 | 0.011 | 2.08 | 0.037 | ** | DTL | 0.015 | 0.007 | 2.22 | 0.026 | ** |
| DT | −0.017 | 0.011 | −1.64 | 0.101 | | DT | −0.018 | 0.006 | −2.8 | 0.005 | *** |
| CP | 0.003 | 0.003 | 0.86 | 0.39 | | CP | 0.004 | 0.002 | 1.87 | 0.062 | * |
| CA | 0.017 | 0.005 | 3.45 | 0.001 | *** | CA | −0.001 | 0.003 | −0.2 | 0.842 | |
| TAX | 0.545 | 0.201 | 2.7 | 0.007 | *** | TAX | −0.223 | 0.123 | −1.81 | 0.071 | * |
| Re | 0.105 | 0.05 | 2.09 | 0.037 | ** | Re | −0.054 | 0.031 | −1.76 | 0.079 | * |
| CT | 0.002 | 0.005 | 0.37 | 0.712 | | CT | 0.015 | 0.003 | 4.91 | 0 | *** |
| PB | −0.126 | 0.05 | −2.51 | 0.012 | ** | PB | 0.039 | 0.031 | 1.28 | 0.201 | |
| PrB | 0.054 | 0.051 | 1.07 | 0.287 | | PrB | −0.053 | 0.031 | −1.69 | 0.092 | * |
| SAL | −401.387 | 562.255 | −0.71 | 0.475 | | SAL | −533.426 | 344.224 | −1.55 | 0.122 | |
| ROE | −6790.91 | 17,720.01 | −0.38 | 0.702 | | ROE | 5241.89 | 10,848.56 | 0.48 | 0.629 | |
| ROA | −284,917 | 959,164.2 | −0.3 | 0.766 | | ROA | 896,560.5 | 587,220.1 | 1.53 | 0.127 | |
| S | 11,106.18 | 48,933.29 | 0.23 | 0.82 | | S | −31,036.2 | 29,957.97 | −1.04 | 0.3 | |
| IF | 212.142 | 616,461 | 0 | 1 | | IF | −443,489 | 377,410.1 | −1.18 | 0.24 | |
| 2011 | 313,097.4 | 408,497.3 | 0.77 | 0.444 | | 2011 | 207,985.2 | 250,090.5 | 0.83 | 0.406 | |
| 2012 | −614,578 | 410,092.8 | −1.5 | 0.134 | | 2012 | −349,674 | 251,067.2 | −1.39 | 0.164 | |
| 2013 | −954,219 | 410,667.2 | −2.32 | 0.02 | ** | 2013 | 135,943.8 | 251,418.9 | 0.54 | 0.589 | |
| 2014 | −567,735 | 411,878.7 | −1.38 | 0.168 | | 2014 | −61,692.3 | 252,160.6 | −0.24 | 0.807 | |
| 2015 | 1,394,940 | 413,768.3 | 3.37 | 0.001 | *** | 2015 | 587,376.8 | 253,317.5 | 2.32 | 0.021 | ** |
| 2016 | −893,451 | 413,159.2 | −2.16 | 0.031 | ** | 2016 | −426,079 | 252,944.6 | −1.68 | 0.092 | * |
| 2017 | −1,316,719 | 417,333.1 | −3.16 | 0.002 | *** | 2017 | −548,530 | 255,499.9 | −2.15 | 0.032 | ** |
| 2018 | −1,370,212 | 418,613.1 | −3.27 | 0.001 | *** | 2018 | −491,738 | 256,283.6 | −1.92 | 0.055 | * |
| 2019 | −1,178,775 | 421,605.6 | −2.8 | 0.005 | *** | 2019 | −440,991 | 258,115.7 | −1.71 | 0.088 | * |
| 2020 | −1,205,464 | 422,664.4 | −2.85 | 0.004 | *** | 2020 | −278,980 | 258,763.9 | −1.08 | 0.281 | |
| 2021 | −132,225 | 432,588.7 | −0.31 | 0.76 | | 2021 | −411,495 | 264,839.8 | −1.55 | 0.121 | |
| Constant | 44,115.06 | 411,880.7 | 0.11 | 0.915 | | Constant | 580,274.3 | 252,161.9 | 2.3 | 0.022 | ** |

| Mean dependent var | 3,471,659.876 | SD dependent var | 6 × 10⁶ | Mean dependent var | 2,112,319.416 | SD dependent var | 3,744,237 |
|---|---|---|---|---|---|---|---|
| R-squared | 0.545 | Number of obs | 1200 | R-squared | 0.452 | Number of obs | 1200 |
| F-test | 41.258 | Prob > F | 0 | F-test | 28.481 | Prob > F | 0 |
| Akaike crit. (AIC) | 39,012.706 | Bayesian crit. (BIC) | 39,176 | Akaike crit. (AIC) | 37,835.116 | Bayesian crit. (BIC) | 37,998 |

*** $p < 0.01$, ** $p < 0.05$, * $p < 0.1$. Source: compiled by the authors with the help of STATA software V13.

From Table 4, it can be observed that there is a significant seasonal influence both in the case of the model regarding the fiscal effect (VATC) and in the case of the model regarding profitability functions (VATD). The most significant influences are recorded in the period 2015–2019, and it can be noted that periods of uncertainty influence the increase in the heterogeneity of the models (Figure 4). One may observe that changes in fiscal policy such as changes in the VAT regime (2010), the restriction of the right to deduct cars and fuels (2011), the formation of the fiscal group (2012), VAT liability upon collection (2012),

the reduction of the tax base in case of bankruptcy (2013), the transformation of the VAT cash accounting system into an optional system (2013), and tax rate changes in 2017 have an impact on the significance of the estimated model based on time-fixed effects presented in Table 4.

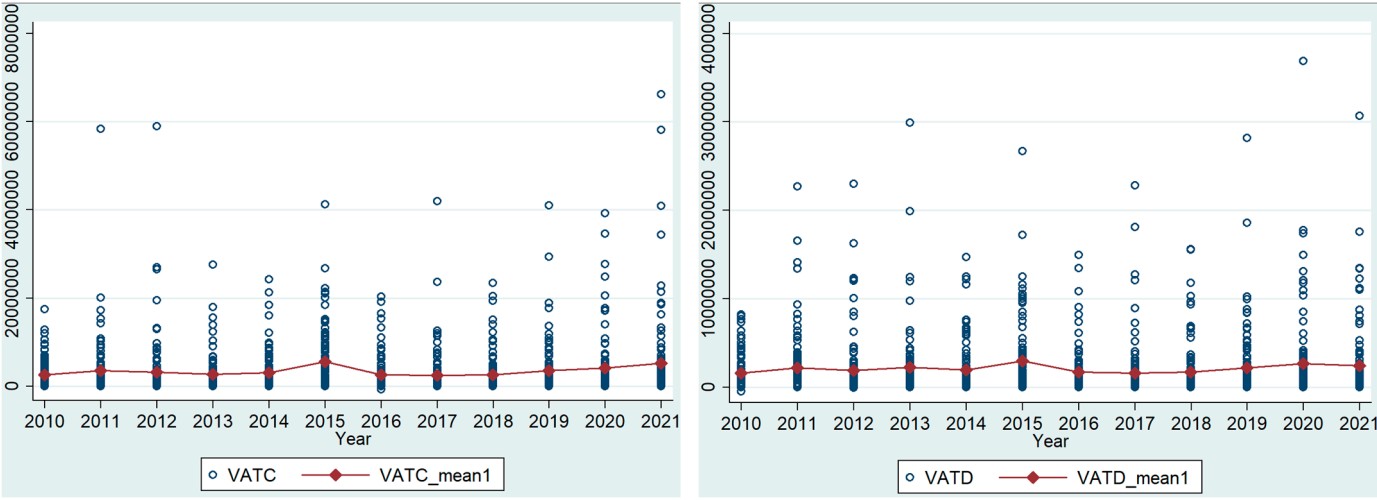

**Figure 4.** Fixed effects: heterogeneity across entities (deductible VAT vs. collected VAT). Source: compiled by the authors with the help of STATA software V13.

The model results for the year 2010 reflect the high homogeneity of data for both VAT correlation models with financial indicators. The correlation value of the deductible VAT indicator, determined through the adjusted R-squared coefficient, is 77.2%. The correlation value of the collected VAT indicator, determined by the adjusted R-squared coefficient, is 71.9%. One may observe a higher variation in the deductible VAT model for the year 2010, which signifies a tendency of the managers in the construction sector to avoid the VAT fiscal vector (see Figure 4). The 2011 correlation value of the deductible VAT indicator, determined through the adjusted R-squared coefficient of determination, is 89.9%. The correlation value of the collected VAT indicator, determined by the adjusted R-squared coefficient of determination, is 85%. Higher variability is observed for the deductible VAT model in 2011 (Time-fixed effects VATC 313097.4, *p*-value 0.444 versus VATD 207985.2, *p*-value 0.406), indicating a tendency of managers in the construction sector to avoid the VAT tax vector. The level of the F function in the case of deductible VAT is 44.567, superior to the level of the F function in the case of collected VAT, which is 28.682 (this aspect shows that the overall significance of the independent variables is better characterized in the case of deductible VAT). The Durbin–Watson coefficient indicates a shift of asymmetry towards the left. In the case of deductible VAT and collected VAT indicators, the 2012 correlation values determined through the adjusted R-squared coefficient of determination are 86.5% and 91.3%, respectively. In the year 2012, a higher variation in the model is observed in the case of collected VAT, which signifies a tightening of the VAT regime in terms of the regulation of collection procedures. The level of the F function in the case of deductible VAT is 32.5, lower than the level of the F function in the case of collected VAT, which is 52.137 (the overall significance of independent variables better characterizing the collected VAT for the year 2012). High data homogeneity for both VAT correlation models with financial indicators is also found in the results of the model applied in 2013 (Time-fixed effects VATC −954219, *p*-value 0.02 versus VATD 135943.8, *p*-value 0.589). The correlation value of the deductible VAT indicator determined through the adjusted R-squared coefficient is 74.6%, while the correlation value of the collected VAT indicator is 86.1%. A higher variability of the model for the collected VAT is observed for the year 2013, representing a tightening of the VAT regime in the regulation of collection procedures. The level of the F function in the case of deductible VAT is 15.275, lower than the level of the F function in the case of

collected VAT, which is 30.983, indicating an overall significance of independent variables favoring collected VAT. We may observe that for the year 2014, there is higher variation in the model for the collected VAT. We find ourselves in the same situation as in previous years when it comes to tightening the VAT regime in the regulation of collection procedures. The level of the F function for deductible VAT is 32.387, lower than the level of the F function for collected VAT, which is 35.921. Like the previous year, for both the deductible VAT and collected VAT indicators, the Durbin–Watson coefficient indicates an asymmetrical shift towards the right.

The year 2015 shows a high level of homogeneity of data for both correlation models of VAT with financial indicators (Time-fixed effects VATC 1394940, *p*-value 0.001 versus VATD 587376.8, *p*-value 0.021). The correlation value of the deductible VAT indicator is 87%, while the correlation value of the collected VAT indicator is 91.9%, both values being determined through the adjusted R-squared coefficient. In 2015, the Durbin–Watson coefficient indicates a shift of asymmetry towards the right in the case of deductible VAT, while in the case of collected VAT, it indicates a symmetrical distribution.

Both VAT correlation models with financial indicators exhibit high data homogeneity in the year 2016 (Time-fixed effects VATC −893451, *p*-value 0.031 versus VATD −426079, *p*-value 0.092). In 2013 and 2016, Romania launched and expanded its own reverse charge mechanism (European Commission 2023b). The correlation value for the deductible VAT indicator is 84.3%, while the correlation value for the collected VAT indicator is 84.5%. The same trend of tightening the fiscal regime persists in 2016, where we continue to observe a higher variation in the model for the collected VAT. The level of the F function for the deductible VAT is 28.71, lower than the F-function level for the collected VAT of 29.022.

The model results reflect the high homogeneity of data for both VAT correlation models with financial indicators in the year 2017 (see Table 4); the correlation value for the deductible VAT indicator is 91.4%, and for the collected VAT indicator 91%. A higher variation is observed for the deductible VAT model in 2017, indicating a tendency for managers in the construction sector to avoid the VAT tax vector, after the promulgation of the law on the deed in lieu of certain real estate (Romanian Parliament 2016). The F-function level for the deductible VAT is 56.096, which is higher than the F-function level for the collected VAT, which stands at 53.337. For both deductible VAT indicators as well as the collected VAT, the Durbin–Watson coefficient indicates an asymmetrical shift towards the left.

In 2018, from the analysis conducted, we find a high uniformity of data for both VAT correlation models with financial indicators (Time-fixed effects VATC −1370212, *p*-value 0.001 versus VATD −491738, *p*-value 0.055). The correlation value for the analyzed indicators is 69.7% for deductible VAT and 74.5% for collected VAT, the values being determined through the adjusted R-squared coefficient of determination. Collected VAT presents a higher level of variation in the model, indicating a worsening of the VAT collection regime. Romania implemented a mandatory split payment scheme for taxable individuals and public institutions with tax arrears or under insolvency procedures between 2018 and 2020 (European Commission 2023b). The F function for deductible VAT has a value of 12.878, lower than the F-function level for collected VAT, which is 16.062. For both deductible VAT and collected VAT, the Durbin–Watson coefficient indicates the symmetry of distribution.

The high data homogeneity for both VAT correlation models with financial indicators is maintained even in the results of the model applied in 2019 (see Table 4), with a correlation value of 87% for the deductible VAT indicator and 89.5% for the collected VAT indicator. There is a continuation of a high level of tax collection in 2019; the collected VAT model shows a higher variation. The F-function level in the case of deductible VAT is 35.621, lower than the F-function level in the case of collected VAT, which is 44.774. For both analyzed indicators, deductible and collected VAT, the Durbin–Watson coefficient indicates a symmetry of the distribution.

The level of data homogeneity for both VAT correlation models with financial indicators is maintained for the results obtained from applying the model for the year 2020. The correlation value for the deductible VAT indicator is 90.5%, and 89.3% for the collected VAT indicator. A higher variation is found for the collected VAT model, maintaining the same VAT collection regime through collection procedures. The deductible VAT exhibits an F-function level of 42.658, superior to the F-function level in the case of collected VAT, which stands at 37.437. For the deductible VAT, the Durbin–Watson coefficient indicates an asymmetrical shift towards the right. In the case of collected VAT, the Durbin–Watson coefficient indicates a symmetry in the distribution.

In the case of the model results applied for the year 2021, the high homogeneity of the data is maintained for both correlation models of VAT with the financial indicators, with a correlation value of the deductible VAT indicator at 91.9%, while the correlation value of the collected VAT indicator stands at a percentage of 92.6%. Also, for the year 2021, the collected VAT model presents a higher degree of variation, as the level of the F function in the case of deductible VAT is 47.072, which is lower than the level of the F function in the case of collected VAT, which is 51.754. The results of the analysis conducted reveal that major differences in the correlation model of VAT with changes in financial indicators occurred during the period 2010–2017, a period marked by changes in the VAT regime (2010), limitation of deductibility, the restriction of the right to deduct cars and fuels (2011), the formation of the fiscal group (2012), VAT liability upon collection (2012), the reduction of the tax base in case of bankruptcy (2013), the transformation of the VAT cash accounting system into an optional system (2013), the reduction of the VAT rate to 9% for tourism (2015), and the establishment of the tax code applicable from 01.01.2016. In 2017, the 19% threshold which had been in place before 2010 was reinstated. Other major differences in the correlation models of deductible VAT and collected VAT with financial indicators were observed in 2021 (Emergency Ordinance 19/2021 regarding the modification of the VAT cash threshold to RON 4,500,000). In conclusion, the correlation model reflects the three major thresholds of changes in the VAT regime in the years 2010, 2016, and 2017, thus confirming hypothesis H4, namely that analytical economic and financial indicators tend to follow the trend of direct correlation with VAT fiscal policy, meaning that in Romania, the level of regulation of the VAT fiscal policy is optimal.

To test the representativeness level of the model using the one-tailed critical probability test, it has been demonstrated (Table 5) that the coefficient values of the F function's Sig. do not exceed the significance threshold of fixed errors in the seasonal variant ($\alpha < 0.05$), allowing for the validation of the alternative hypothesis and the rejection of the null hypothesis, thereby validating the bivariate correlation model of VAT (deductible VAT vs. collected VAT). Furthermore, through the ANOVA test, it was demonstrated that the sum of the model's regression squares in the total distribution of regression squares favor deductible VAT mainly in the years 2010, 2011, 2017, and 2020, which coincides with the onset of economic crises or changes in the VAT fiscal regime, aspects that we consider significant in developing a sustainable fiscal instrument whose structure can be defined according to the following lemma:

**Table 5.** ANOVA.

| Anova | | Sum Square | df | Mean Square | F | *p* Value |
|---|---|---|---|---|---|---|
| TVAC | Total | 46,191,762,693,719,800 | 1199 | | | 0.00 |
| | Error | 10,712,804,112,352,200 | 1179 | 9,086,347,847,627 | | 0.00 |
| | Regression | 35,478,958,581,367,500 | 20 | 1,773,947,929,068,370 | 195.232 | 0.00 |
| TVAD | Total | 16,809,151,914,893,000 | 1199 | | | 0.00 |
| | Error | 4,315,520,612,405,130 | 1179 | 3,660,322,826,467 | | 0.00 |
| | Regression | 12,493,631,302,487,900 | 20 | 624,681,565,124,398 | 170.663 | 0.00 |

Source: compiled by the authors using SmartPLS4 statistical software.

The sustainable fiscal regime that allows for the economic development of the private sector is characterized by protecting direct investments in the economy with the application of limits to safeguard these investments and by protecting social rights with limits applied to entities that effectively contribute to the sustainability of the social economy. Moreover, possible changes in the VAT fiscal regime to achieve sustainability should occur as seldom as possible, ensuring the necessary and sufficient predictability for the expected degree of voluntary compliance.

By establishing the correlation between fixed asset indicators and the deductible level of VAT in terms of the dependency of deductible VAT variation on the variation of the analyzed indicator, the direct correlation reflects an upward trend, with the minimum value of the correlation of 0.003 having been reached in the year 2012, while the maximum value of the Pearson correlation of 0.546 was determined for the year 2013. The Pearson correlation for 2021 was estimated at a value of 0.378 using statistical correlation tests. This signifies that the variation in deductible VAT recorded in the construction sector for the sample of 100 selected firms is directly influenced to the extent of 37.8% by the variation in fixed assets. In other words, an increase of RON 1 in fixed assets induces an increase of RON 0.378 in deductible VAT. The analysis of the correlation between fixed asset indicators and the collected level of VAT in terms of the dependency of collected VAT variation on the variation of the analyzed indicator indicates an upward trend in the direct correlation. The minimum threshold of the correlation was reached in the year 2010 (−0.006), while the maximum threshold was reached in the year 2014 (0.501). In the final year of the analysis, namely 2021, it was demonstrated through the Pearson correlation test that the dynamics of collected VAT are influenced to a percentage of 42% by the dynamics of fixed assets. In other words, an increase of RON 1 in fixed assets induces an increase of RON 0.424 in collected VAT. Establishing the correlation between receivables indicators and the deductible level of the VAT in terms of the dependency of deductible VAT variation, the variations in the analyzed indicators reflect an upward trend in the direct correlation. For the minimum correlation, the value reached 0.469, attained in the year 2013, while the maximum value of the Pearson correlation of 0.873 was determined for the year 2020. Through statistical correlation tests for the year 2021, the Pearson correlation was estimated at a value of 0.804, signifying that the variation in deductible VAT recorded in the construction sector for the sample of 100 selected firms is directly influenced to the extent of 80.4% by the variation in receivables, meaning that an increase of RON 1 in receivables induces an increase of RON 0.804 in deductible VAT.

The correlation between receivables indicators and the collected VAT level in terms of the dependency of collected VAT variation on the variation of the analyzed indicator reflects a decreasing trend in the inverse correlation. The minimum value of the correlation of 0.536 was reached in 2015, while the maximum value of the Pearson correlation of 0.841 was determined for the year 2019. In 2021, the Pearson correlation was estimated at a value of 0.779 by means of the statistical correlation test. This signifies that the variation in collected VAT recorded in the construction sector for the sample of 100 selected firms is directly proportionally influenced by 77.9% of the variation in receivables. In other words, an increase of RON 1 in receivables induces an increase of RON 0.779 in collected VAT. The correlation between current assets indicators and the deductible level of VAT in terms of the dependency of deductible VAT variation on the variation of the analyzed indicator reflects an ascending trend in the direct correlation. The minimum value of the correlation of 0.584 was reached in 2013, while the maximum value of the Pearson correlation of 0.886 was determined for the year 2020. In 2021, the Pearson correlation was estimated at a value of 0.827 by means of the statistical correlation test. This signifies that the variation in deductible VAT recorded in the construction sector for the sample of 100 selected firms is directly influenced in proportion by 82.7% of the variation in current assets. In other words, an increase of RON 1 in current assets induces an increase of RON 0.827 in deductible VAT. The correlation between current asset indicators and the collected VAT level in terms of the dependency of collected VAT variation on the variation of the analyzed indicator reflects

an ascending trend in the direct correlation. The minimum value of the correlation of 0.704 was reached in 2017, while the maximum value of the Pearson correlation was determined for the year 2021, through the statistical correlation test, at a value of 0.879. This signifies that the variation in collected VAT recorded in the construction sector for the sample of 100 selected firms is directly influenced in proportion by 87.9% of the variation in current assets. In other words, an increase of RON 1 in current assets induces an increase of RON 0.879 in collected VAT.

The correlational analysis of total assets and deductible VAT indicates the dependence of deductible VAT variation on the variation of the analyzed indicator, with the direct correlation showing an upward trend. The minimum value of the correlation was 0.145 in the year 2011, while the maximum value of the Pearson correlation of 0.899 was determined for the year 2020. Through the statistical test in 2021, the Pearson correlation was estimated at 0.837. The variation of deductible VAT recorded in the construction sector for the sample of 100 selected firms is directly influenced by 83.7% due to the total asset variation. An increase of RON 1 in total assets induces an increase of RON 0.837 in deductible VAT. By establishing the correlation between the total assets indicator and the collected VAT indicator, a dependency of collected VAT variation on the variation of the analyzed indicator is observed. The direct correlation shows an upward trend, ranging from a minimum in 2011 with a correlation value of 0.141 to a maximum Pearson correlation of 0.893 determined for the year 2021, indicating that the variation of collected VAT in the construction sector for the sample of 100 selected firms is directly influenced by 89.3% of the total asset variation. The analysis of the correlational distribution between the indicator of other liabilities, including tax and social security liabilities, and the level of VAT deduction, demonstrates an upward trend in Pearson's direct correlation because of the dependence of deductible VAT variation on the variation of the analyzed indicator, with a minimum value of 0.306 in 2013 and a maximum value of 0.799 reached in 2017. In 2021, the Pearson correlation was estimated through the statistical correlation test at a value of 0.658, where an increase of RON 1 in other short-term liabilities, including tax and social security liabilities, induces an increase of RON 0.658 in deductible VAT. Concerning the correlation between the indicators of other short-term liabilities, including tax and social security liabilities, and the level of collected VAT, through the dependence of collected VAT variation on the variation of the analyzed indicator, a minimum value of the Pearson correlation of 0.38 was determined in 2012, and a maximum value of 0.839 was determined for 2017. In the year 2021, the Pearson correlation was estimated through the statistical test at a value of 0.801, indicating that the variation of collected VAT recorded in the construction sector for the sample of selected firms is directly influenced by 80.1% of the variation of other short-term liabilities, including tax and social security liabilities.

The correlation between the total short-term debt indicators and the deductible VAT rate in terms of the dependency of deductible VAT variation on the variation of the analyzed indicator reflects an upward trend in direct correlation. The minimum correlation value of 0.55 was reached in 2013, while the maximum value of the Pearson correlation, 0.823, was determined for the year 2011. In 2021, the Pearson correlation was estimated through the statistical correlation test at a value of 0.69. This signifies that the variation in deductible VAT recorded in the construction sector for the selected sample of 100 firms is directly influenced by 69% of the total short-term debt variation. In other words, an increase of one unit in total short-term debts induces an increase of 0.69 units in deductible VAT. The analysis of correlation between the total short-term debt indicator and the collected VAT level under the conditions of dependency regarding the VAT collected variation based on the analyzed indicator demonstrates how the Pearson correlation registers a minimum correlation value of 0.582 in 2017 and a maximum value of 0.827 determined for the year 2019. For the year 2021, through statistical testing, the Pearson correlation reached a value of 0.824, showing that an increase of one unit in total short-term debts induces an increase of 0.824 units in collected VAT. The correlation coefficient of the other debts indicator, including long-term tax and social security debts, with deductible VAT,

respecting the dependency of the variation of deductible VAT on the analyzed indicator, demonstrates a decreasing trend in direct correlation. This trend exhibits a minimum value of 0.232 in the year 2021 and a maximum value of 0.618 determined for the year 2017. The Pearson correlation was estimated in 2021 at a value of 0.232, indicating that the variation in deductible VAT recorded in the construction sector for the analyzed firms is directly influenced by 23.2% of the variation in the other debts indicator, including long-term tax and social security debts. The correlation coefficient of the other debts indicator, including long-term tax and social security debts, with the collected VAT indicator, respecting the dependency of the variation of collected VAT on the analyzed indicator, exhibits a downward trend for direct correlation. The minimum correlation value was 0.248 in 2019, while the maximum correlation value determined for 2017 was 0.568. Through the statistical test in 2021, the Pearson correlation was estimated at a value of 0.355, indicating that the change in the collected VAT value in the analyzed sample is directly influenced by 35.5% of the variation in the other debts indicator, including long-term tax and social security debts. Specifically, an increase of RON 1 in the other debts indicator, including long-term tax and social security debts, leads to an increase of RON 0.355 in collected VAT.

The calculation of the correlation between the indicator of total long-term debt and the level of deductible VAT, given the dependence of deductible VAT on the variation of the analyzed indicator, indicates an upward trend in the direct correlation. The minimum value was recorded at 0.195 in 2011, while the maximum value of the Pearson correlation determined for the year 2017 reached 0.686. In 2021, the correlation was estimated at 0.311. This situation reveals that the variation in deductible VAT observed in the construction sector for the selected sample of companies is directly influenced by 31.1% of the variation in total long-term debts. Regarding the correlation calculation between the indicators of total long-term debts and collected VAT, while considering the dependence of the collected VAT variation on the variation of the analyzed indicator, it shows a decreasing trend. The minimum correlation value of 0.167 was reached in 2011, while the maximum value was 0.76 in 2017. In 2021, the Pearson correlation was estimated through a statistical correlation test at 0.326. Specifically, a RON 1 increase in total long-term debts induces a RON 0.326 increase in collected VAT. The correlation between total debts and the deductible VAT level concerning the dependence of the deductible VAT variation on the variation of the analyzed indicator reflects an upward trend in the direct correlation. The minimum correlation value of 0.569 was recorded in 2013, while the maximum Pearson correlation of 0.83 was determined for 2019. In 2021, the Pearson correlation was estimated at 0.703. This signifies that the variation in deductible VAT observed in the construction sector for the sample of 100 selected companies is directly influenced by 70.3% of the variation in total debts. In other words, a RON 1 increase in total debts induces a RON 0.703 increase in deductible VAT.

The correlation between total liabilities and the collected level of VAT, in terms of the dependence of VAT variation on the variation of the analyzed indicator, reflects a descending trend of the inverse correlation. The minimum value of the correlation, 0.629, was reached in 2010, while the maximum value of the Pearson correlation, 0.826, was determined for the year 2021. In 2021, the Pearson correlation was estimated through the statistical correlation test at a value of 0.826. This signifies that the variation in collected VAT recorded in the construction sector for the sample of 100 selected firms is directly influenced to an extent of 82.6% by the variation in total liabilities. In other words, an increase of RON 1 in total liabilities induces an increase of RON 0.826 in collected VAT. The correlative analysis between equity indicators and deductible VAT, under the same dependency conditions, indicates a descending trend of the direct correlation, with a minimum correlation value of −0.005 in 2021, and a maximum correlation value of 0.403 in 2015. In 2021, the Pearson correlation was estimated through the statistical correlation test at a value of −0.005; in such conditions, the variation in deductible VAT recorded in the construction sector for the selected sample is indirectly influenced to the extent of −0.5% by the variation in equity.

In the case of equity indicators and the level of collected VAT, the correlational analysis conducted in terms of the dependency of collected VAT on the variation of the analyzed indicator shows a decrease in the direct correlation. The minimum correlation value is −0.008, reached in 2019, while the maximum correlation value was 0.399, determined in 2015. In 2021, the Pearson correlation was estimated at −0.147. A decrease of RON 1 in equity induces an increase of RON 0.147 in collected VAT. The calculation of the correlation index between turnover indicators and the deductible level of VAT in terms of the dependency of deductible VAT variation on the variation of the analyzed indicator expresses a descending trend of the inverse correlation. The minimum correlation value is 0.605 in 2017, and the maximum value of 0.918 was determined for 2015. For 2021, the Pearson correlation was 0.89. This signifies that the variation of deductible VAT recorded in the construction sector for the sample of 100 selected firms is directly influenced by 89% of turnover variation. Regarding turnover indicators and the level of collected VAT, the correlation index calculation indicates a decreasing evolution of the inverse correlation. The minimum correlation value was 0.516 in 2017, and the maximum value was 0.937 determined for 2021. Thus, the variation of collected VAT recorded in the construction sector for the selected firms is directly influenced by 93.7% of turnover variation.

For indicators related to expenses on other taxes, fees, contributions, and deductible VAT, the correlational analysis under conditions of dependency of deductible VAT variation in relation to the variation of the analyzed indicator shows a decreasing trend in direct correlation. The minimum value of the correlation is 0.346 in the year 2017, while the maximum value of the Pearson correlation is 0.566 and was determined for the year 2012. However, in the year 2021, the Pearson correlation was estimated at a value of 0.497 through the statistical correlation test. This aspect indicates that the variation of deductible VAT is directly influenced by 49.7% of the variation in expenses on other taxes, fees, and contributions. Regarding the correlation between the indicator of expenses on other taxes, fees, contributions, and the collected VAT, a decreasing trend in direct correlation is observed, with a minimum value of 0.256 in the year 2010 and a maximum value of the Pearson correlation of 0.669 achieved in the year 2014. In the year 2021, the Pearson correlation was estimated at a value of 0.525 through the statistical correlation test. Broadly speaking, an increase of one unit in expenses on other taxes, fees, and contributions generates an increase of 0.525 units in collected VAT.

In terms of the dependence of deductible VAT variation on the variation of the operating result indicator, the correlational analysis between these indicators reflects a downward trend in direct correlation. It records a minimum correlation value of 0.041 in the year 2018 and a maximum Pearson correlation value of 0.51 determined in the year 2013. For the year 2021, the Pearson correlation was estimated at a value of 0.191. Consequently, the variation of deductible VAT recorded in the construction sector for the studied sample is directly influenced to the extent of 19.1% by the variation in the operating result. The correlation of collected VAT variation with the variation of the operating result indicator indicates a descending trend in direct correlation. The minimum correlation value of 0.105 was reached in the year 2018, while the maximum Pearson correlation value of 0.61 was determined for the year 2014. For the year 2021, the Pearson correlation was estimated at a value of 0.239. Thus, an increase of RON 1 in the operating result generates an increase of RON 0.239 in collected VAT.

The correlation between total income indicators and the deductible level of the VAT rate in terms of the dependence of deductible VAT variation on the variation of the analyzed indicator reflects an upward trend of direct correlation. The minimum correlation value of 0.638 was reached in the year 2017, while the maximum value of the Pearson correlation of 0.924 was determined for the year 2015. In the year 2021, the Pearson correlation was estimated through the statistical correlation test at a value of 0.881. This signifies that the variation in deductible VAT recorded in the construction sector for the sample of 100 selected firms is directly influenced by 88.1% of the variation in total income. In other words, a RON 1 increase in total income induces a RON 0.881 increase in deductible VAT.

The correlation between total income indicators and the level of collected VAT in terms of the dependence of collected VAT variation on the variation of the analyzed indicator reflects a downward trend of inverse correlation. The minimum correlation value of 0.533 was reached in the year 2017, while the maximum value of the Pearson correlation of 0.944 was determined for the year 2021. In the year 2021, the Pearson correlation was estimated through the statistical correlation test at a value of 0.944. This signifies that the variation in collected VAT recorded in the construction sector for the sample of 100 selected firms is directly influenced by 94.4% of the variation in total income. In other words, a RON 1 increase in total income induces a RON 0.944 increase in collected VAT.

Correlative analysis of total expenditure indicators and the deductible level of the VAT rate in terms of the dependence of deductible VAT variation on the variation of the abovementioned indicator generates a direct ascending correlation. The minimum value of the correlation was 0.67 in 2017, and a maximum value of 0.915 was observed in 2015. In the year 2021, through the statistical correlation test, the Pearson index had an estimated value of 0.882. Overall, the variation of deductible VAT recorded in the construction sector for the sample of 100 selected companies is directly influenced by 88.2% of the total expenditure variation. Regarding the total expenditure indicators and the collected VAT rate, correlative analysis, respecting the dependency of collected VAT variation on the variation of the total expenditure indicator, indicates an increase in direct correlation. The minimum correlation value was 0.568 in 2017, and the maximum correlation value of 0.943 was determined for the year 2021. Therefore, an increase of RON 1 in total expenditures induces an increase of RON 0.943 in collected VAT. For the gross result and deductible VAT indicators, the Pearson correlation calculation indicated a descending trend of direct correlation. The minimum correlation value was 0.012 in 2011, while the maximum correlation value was 0.461 and was determined for the year 2013. For the year 2021, the Pearson correlation level was estimated at a value of 0.151, indicating that the variation of deductible VAT recorded in the construction sector for the sample of selected firms is directly influenced by 15.1% of the gross result variation.

In the case of correlative analysis for the gross result and collected VAT indicators, a descending trend of inverse correlation is maintained. The minimum correlation value of 0.098 was reached in 2018, while the maximum correlation value of 0.567 was determined for the year 2014. In 2021, the Pearson correlation was estimated at a value of 0.187, so an increase of RON 1 in gross result induces an increase of RON 0.187 in collected VAT. The Pearson correlation index calculated for gross profit and deductible VAT indicators shows an ascending trend of direct correlation. The minimum correlation value was 0.14 reached in 2017, while the maximum correlation value of 0.605 was determined for the year 2019. By 2021, we recorded a Pearson correlation value of 0.551. This signifies that the variation of deductible VAT recorded in the construction sector for the sample of 100 selected firms is directly influenced by 55.1% of the gross profit variation.

For gross profit and collected VAT indicators, the calculated Pearson correlation shows a descending trend of inverse correlation. The minimum correlation value of 0.204 was reached in 2017, and the maximum value of 0.632 was determined for the year 2019. In 2021, the Pearson correlation was estimated at a value of 0.559. The variation of collected VAT recorded in the construction sector for the analyzed sample is directly influenced by 55.9% of the gross profit variation, or in other words, a RON 1 increase in gross profit leads to a RON 0.559 increase in collected VAT. For the gross loss and deductible VAT indicators, the Pearson correlation, given the dependency of deductible VAT variation on gross loss variation, shows an ascending trend of direct correlation, registering a minimum correlation value of 0.014 in the year 2015 and a maximum correlation value of 0.483 in the year 2017. In 2021, the Pearson correlation was estimated at a value of 0.17. Thus, the variation in deductible VAT recorded in the construction sector for the analyzed sample is directly influenced by 17% of the gross loss variation.

The correlative analysis of the gross loss and collected VAT indicators shows an increase in inverse correlation. The minimum correlation value is 0.084 in 2015, and the

maximum correlation value is 0.528, determined for the year 2017. In 2021, the Pearson correlation was estimated at a value of 0.131, and the variation in collected VAT for the analyzed sample is directly influenced by 13.1% of the gross loss variation. An increase of one unit in gross loss generates an increase of 0.131 units in collected VAT. All these observations regarding the correlation of analytical economic–financial indicators with deductible VAT and collected VAT confirm hypothesis H4, namely "that analytical economic–financial indicators tend to follow a direct correlation trend with VAT fiscal policy, indicating that in Romania, the level of regulation of VAT fiscal policy is optimal".

In terms of the dependence of deductible VAT variation on the variation in the number of employees, correlational analysis shows an ascending trend in the direct correlation. The minimum value of 0.2 was reached in 2011, and the maximum value of 0.555 was determined for the year 2021. Following the recorded values, the variation in deductible VAT in the construction sector for the analyzed sample is directly influenced by 55.5% of the variation in the number of employees; an increase of one employee leads to an increase of RON 0.555 in deductible VAT. In the correlational analysis between the indicators of the number of employees and collected VAT, there is an upward trend in the direct correlation, with a minimum correlation value of 0.169 recorded in 2010, while the maximum correlation value of 0.563 was determined in 2020. The Pearson correlation for the year 2021 was estimated at 0.538, indicating that the recorded variation in collected VAT in the construction sector for the selected sample is directly influenced by 53.8% of the variation in the number of employees; an increase of one employee leads to an increase of RON 0.538 in collected VAT.

The correlation between Return on Equity (ROE) indicators and deductible VAT in terms of the dependence of deductible VAT variation on the variation in the analyzed indicator reflects a downward trend in the direct correlation. The minimum correlation value of $-0.009$ was reached in 2016, while the maximum Pearson correlation of 0.224 was determined for the year 2011. In 2021, the Pearson correlation was estimated through the correlation test at $-0.04$. This implies that the recorded variation in deductible VAT in the construction sector for the selected sample of 100 firms is indirectly influenced by $-4\%$ of the variation in Return on Equity (ROE). In other words, a decrease of 1% in Return on Equity (ROE) leads to an increase of RON 0.04 in deductible VAT. The analysis of the correlation between Return on Equity (ROE) and collected VAT shows an upward trend in the inverse correlation. The minimum correlation value of 0.001 was reached in 2017, while the maximum value of the Pearson correlation of $-0.211$ was determined for the year 2019. In 2021, the Pearson correlation was estimated through the statistical correlation test at a value of $-0.068$. This signifies that the variance in collected VAT observed in the construction sector for the sample of 100 selected firms is indirectly influenced by $-6.8\%$ of the variance in Return on Equity (ROE). In other words, a 1% decrease in Return on Equity (ROE) results in an increase of RON 0.068 in collected VAT. The analysis of the correlation between Return on Assets (ROA) and deductible VAT resulted in an increase in the inverse correlation. The minimum correlation value of $-0.008$ was reached in 2011, and the maximum value was $-0.236$ for the year 2020. In 2021, the Pearson correlation estimated through the statistical test recorded a value of 0.011, indicating that the recorded variation in deductible VAT in the construction sector for the studied sample is directly influenced by 1.1% of the variation in Return on Assets (ROA); a 1% increase in Return on Assets (ROA) leads to an increase of RON 0.011 in deductible VAT. The correlational analysis of the Return on Assets (ROA) and collected VAT indicators, respecting the dependency conditions of the collected VAT variation based on the variation in the analyzed indicator, indicates an increase in the inverse correlation with a minimum value of 0.001 in the year 2010 and a maximum value of $-0.218$ determined in the year 2020. In the year 2021, the Pearson correlation recorded a value of $-0.011$, showing that the variation of collected VAT is indirectly influenced by $-1.1\%$ of the variation in Return on Assets (ROA). A decrease of 1% in Return on Assets (ROA) generates an increase of RON 0.011 in collected VAT.

These aspects go to confirm hypotheses H2 and H3, namely:

**H2.** *The level of profitability expressed synthetically through rates of return on capital and assets varies inversely proportionally to the evolution of collected and deductible VAT, meaning that excessive fiscality is a major hindrance to the sustainable development of firms in Romania.*

**H3.** *Rates of return on equity are less sensitive to the tightening of the VAT regime than rates of return on assets, indicating that in Romania, the tightening of the tax regime leads to a transfer of savings at the expense of the quality offered to customers, while protecting investors' interests remains a priority.*

The Pearson correlation analysis of the indicators of equity solvency and deductible VAT showed an upward trend in the inverse correlation. In the year 2011, the minimum correlation value of 0 was reached, while the maximum value of the Pearson correlation of −0.246 was recorded in the year 2016. In 2021, the Pearson correlation was estimated at a value of −0.092. This signifies that the variation in deductible VAT recorded in the construction sector for the sample of 100 selected firms is indirectly influenced by −9.2% of the variation in equity solvency. In other words, a decrease of 1% in equity solvency induces an increase of RON 0.092 in deductible VAT.

In the case of the indicators of equity solvency and collected VAT, the calculation of the Pearson correlation showed an upward trend in the inverse correlation, with a minimum correlation value of 0.04 in the year 2011 and a maximum value of the Pearson correlation of −0.255 in the year 2016. In the year 2021, the Pearson correlation registered a value of −0.12. The variation of collected VAT for the sample of 100 selected firms is indirectly influenced by −12% of the variation in equity solvency, and a decrease of 1% in equity solvency generates an increase of RON 0.12 in collected VAT.

The analysis of the correlation between the indicators of financial independence and deductible VAT in terms of the dependence of deductible VAT variation on the variation of the analyzed indicator reflects an upward trend in the inverse correlation. The minimum correlation value of −0.103 was reached in 2011, while the maximum Pearson correlation of −0.37 was determined for the year 2020. In 2021, the Pearson correlation was estimated through the correlation test at −0.311. This signifies that the recorded variation in deductible VAT in the construction sector for the selected sample of 100 firms is indirectly influenced by −31.1% of the variation in financial independence. In other words, a decrease of 1% in financial independence leads to an increase of RON 0.311 in deductible VAT.

The correlation between financial independence and the collected VAT in terms of the dependence of collected VAT variation on the variation of the analyzed indicator reflects an upward trend in the inverse correlation. The minimum correlation value of −0.024 was reached in 2011, while the maximum Pearson correlation of −0.395 was determined for the year 2021. In 2021, the Pearson correlation was estimated through the correlation test at −0.395. This indicates that the recorded variation in collected VAT in the construction sector for the selected sample of 100 firms is indirectly influenced by −39.5% of the variation in financial independence. In other words, a decrease of 1% in financial independence leads to an increase of RON 0.395 in collected VAT.

Thus, this goes to confirm hypothesis H1, namely that financial independence and solvency indicators vary inversely proportionally to the evolution of VAT at the branch level, implying that excessive taxation in Romania represents a significant obstacle to the economic development of entities.

The results of the research contribute to the existing body of knowledge, demonstrating that excessive taxation has a negative impact on the sustainable development of businesses. This primarily affects consumer interests through a decline in the quality of services and an increase in prices. On a secondary level, it affects the interests of investors with long-term effects on industry investments and budgetary revenues. Specifically, in the construction sector in Romania over the past decade, this has led to limitations on deductibility, changes in the VAT regime (2010), restrictions on the right to deduct cars and fuels (2011), the formation of fiscal groups (2012), VAT liability upon collection (2012), the reduction of the

tax base in case of bankruptcy (2013), the transformation of the VAT cash accounting system into an optional system (2013), and tax rate changes in 2017. These measures have directly resulted in a continuous increase in construction project prices, a decrease in the sales of built-up areas of properties, a reduction in investments in the construction industry, and a string of insolvencies following the enactment of the law on the surrender of real estate in payment (Romanian Parliament 2016).

## 5. Conclusions

In this paper, a detailed analysis has been conducted regarding the fiscal–budgetary measures adopted by legislative bodies with competencies in the field at the European level and in Romania, as well as the impact of these measures on the business environment in the construction sector.

According to the analysis, it emerged that the indirect tax on VAT contributes significantly to budget revenues, accounting for over 20% of budget revenues in 2021. The analysis revealed that the main functions of the tax consist of revenue collection for the state budget and regulatory aspects concerning the impact on economic development in strategic areas. The most significant effect was observed in Romania in 2017 when, due to reductions in VAT rates for certain economic sectors, substantial economic growth effects were achieved, especially in the consumer economy.

The analysis highlighted that in Romania, VAT policy has been unpredictable, based solely on the momentary interests of the authorities, namely covering the budget deficit and favoring economic growth.

At the European level, Romania ranked fourth in terms of the value of the general VAT rate applied to goods and services traded within the country, being the only EU state whose VAT dynamics from 2010 to 2022 were negative. This involved a decrease in the VAT rate from 24% in 2010 to 19% at present.

The analysis revealed that at the European level, there are people interested in standardizing fiscal policies which have influenced the VAT policy. European directives have been established to regulate VAT fiscal policy, from Directive 2006/112/EC to Directive 2022/542/EC. The effects of changes in VAT policy at the European level were aimed at developing the digital economy, transitioning towards the green economy, protecting the social economy during the pandemic period, and creating a tax environment tailored to the needs of modern society.

Based on numerous structural changes in the VAT regime, it was found that there is a high level of distrust in the business environment regarding the predictability of fiscal policies, hindering the voluntary compliance process.

The analysis of the impact of fiscal–budgetary measures on the business environment in the construction sector highlighted the fact that the VAT collection base showed dynamic growth within the sector, based on statistical reports from a sample of 100 selected construction companies, increasing from RON 11.76 million to RON 29.11 million. This increase resulted from both changes in the VAT fiscal policy and economic growth and inflation.

The collected VAT dynamics followed an upward trend based on taxation, doubling in volume during the analyzed period. Generally, regarding the balance of VAT payments, there was favorable development until the pandemic struck in the construction sector. However, according to the analysis, the pandemic's impact on economic activity and, implicitly, on meeting obligations towards the state budget in terms of VAT payment was disruptive. There was an over 60% increase in outstanding VAT balances owed by construction companies due to the sector's fiscalization through VAT.

Disruptive aspects were also observed during the pandemic regarding cumulative VAT payments, which we consider to represent a significant effort by construction companies experiencing economic recession (during the period 2020–2021) or subject to abrupt changes in VAT regimes due to fiscal policy changes (2015).

Regarding the economic effects of fiscal policy, it was observed that under the impact of excessive fiscalization, the rate of renewal of fixed assets significantly decreased, registering

a 30% depreciation in the sector from 2010 to 2021. Conversely, due to economic instability and a decrease in debt collection, the value of this indicator with an unfavorable impact on sustainable development increased by over 100% during the analyzed period (from RON 30.15 million in 2010 to RON 64.91 million in 2021).

Debts contributed to the increase in current assets, as the increased value of current assets during the analyzed period was RON 73 million, of which RON 24 million was related to debts. As a result of these two economic aspects, a 60% growth trend in the value of total assets was noted, which we consider a vulnerability, as the uncertain current assets' share in the economic balance increased, while the fixed assets' share decreased. The increase in total assets is attributed to national economic growth, coupled with inflation and the devaluation of the national currency due to a consumption-oriented economy. This aspect is also reflected in the exaggerated increase in the deficit.

If the value of assets increased by 60% during the analyzed period, the value of short-term debts increased by 142%; among these debts, tax debts increased more rapidly, by up to 174%.

Similarly, long-term debts increased by 117%, representing another vulnerable aspect regarding the long-term sustainability of construction businesses in Romania. The turnover value increased by 75% during the analyzed period, exceeding the growth in asset value, indicating that companies in the construction sector only partially capitalized on the results of economic activity, with a capitalization rate evaluated through the ratio between asset growth and turnover at 80%. We can say that the capitalization process is not sustainable, accumulating volatile elements such as debts, which in terms of risk and uncertainty can easily transform into uncertain debts. The fiscalization effect on the sector was extremely high, with a 92% increase during the analyzed period, indicating the need for state budget financing and for the state budget to cover the growing budget deficit, solely due to a defective macroeconomic policy.

Consequently, the evolution of the industry's profitability became negative in 2017, with previous setbacks in the years 2012 and 2014. For the construction sector, we consider that the effects of the pandemic did not constitute an obstacle to the economic development of the sector, according to reported results. In 2021, the first signs of recovery in terms of gross loss reduction at the sector level occurred after 2014 when it started to increase consistently, significantly affecting construction activities.

The proposed econometric model for assessing the impact of VAT on the business environment in Romania's construction sector was based on determining annual models of the correlated evolution of deductible and collected VAT. Highly statistically significant values for both models were obtained for the entire analyzed period, observing from the presented histograms that the collected VAT dynamics had a more homogeneous evolution than deductible VAT. These phenomena should evolve in tandem, but there are significant differences in model structure, confirming that fiscalization rules are more predictable for collected VAT, to the detriment of deductible VAT.

As a conclusion of the modeling, it was shown that a sustainable fiscal regime allows for the economic development of the private sector, characterized by protecting direct investments in the economy with the application of ceilings to safeguard these investments and by protecting social rights with the application of ceilings for entities that contribute effectively to social sustainability. Possible changes in the VAT fiscal regime to achieve sustainability should be implemented as seldom as possible to ensure the necessary and sufficient predictability of the expected degree of voluntary compliance.

The limitations of the research consist of the inclusion of only companies with more than 100 employees in the sample, a restrictive aspect from the perspective of the phenomenon of tax evasion, recognized in the specialized literature as being particularly prevalent among small and medium-sized enterprises. The authors aim to expand the research on a future occasion regarding the studied sample, which will be enlarged to include a greater number of companies, encompassing both small and medium-sized ones.

Additionally, the authors will include companies from other European states in the sample to refine the conclusions and extend them to the European space.

**Author Contributions:** Conceptualization: C.E.B., N.B.-M., M.C., I.S., M.L.Z., C.F. and V.M.A.; Software: C.E.B., N.B.-M., M.C., I.S., M.L.Z., C.F. and V.M.A.; Validation: C.E.B., N.B.-M., M.C., I.S., M.L.Z., C.F. and V.M.A.; Formal Analysis: C.E.B., N.B.-M., M.C., I.S., M.L.Z., C.F. and V.M.A.; Investigation: C.E.B., N.B.-M., M.C., I.S., M.L.Z., C.F. and V.M.A.; Resources: C.E.B., N.B.-M., M.C., I.S., M.L.Z., C.F. and V.M.A.; Data Curation: C.E.B., N.B.-M., M.C., I.S., M.L.Z., C.F. and V.M.A.; Writing—Original Draft Preparation: C.E.B., N.B.-M., M.C., I.S., M.L.Z., C.F. and V.M.A.; Writing—Review & Editing: C.E.B., N.B.-M., M.C., I.S., M.L.Z., C.F. and V.M.A.; Visualization: C.E.B., N.B.-M., M.C., I.S., M.L.Z., C.F. and V.M.A.; Supervision: C.E.B., N.B.-M., M.C., I.S., M.L.Z., C.F. and V.M.A.; Project Administration: C.E.B., N.B.-M., M.C., I.S., M.L.Z., C.F. and V.M.A. All authors have read and agreed to the published version of the manuscript.

**Funding:** This research received no external funding.

**Informed Consent Statement:** All authors have consented to the publication of this manuscript.

**Data Availability Statement:** The data that support the findings of this study are available from the corresponding author upon request.

**Conflicts of Interest:** No potential conflicts of interest were reported by the authors. The authors have no relevant financial or non-financial interests to disclose.

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
