# Peer review of "Modelling the Impact of VAT Fiscality on Branch-Level Performance in the Construction Industry—Evidence from Romania"

_economies, doi:10.3390/economies12020030_

Round 1

Reviewer 1 Report

Comments and Suggestions for Authors

The Article titled 'The Modeling of Impact of VAT Fiscality on Branch-Level 2 Performance in the Construction Industry. Evidence 3 from Romania' is analysing the evolution of VAT fiscality in Romania and its impact on the performance in the construction sector. The main contribution and strength of the paper is in the proposed econometric model for assessing the impact of VAT on the business  environment in Romania's construction sector based on determining annual models  of the correlated evolution of deductible and collected VAT. The research concludes that possible changes in the VAT fiscal regime to achieve sustainability should have as little frequency as possible to ensure the necessary and sufficient predictability for the expected degree of voluntary compliance.

The research topic under investigation is current, the manuscript follows the proposed structure of the journal but some modifications are required. These recommendations are intended to bolster the scientific rigor, clarity, and overall quality of the document.

In the Abstract, it is meaningful to present the results of own research.

The relevance of the manuscript should be explained by appropriate references to define main research questions especially in Introduction where no cited references (mostly recent publications within the last 5 years) were used (just one source). By briefly presenting the most important and relevant references of the review topic the gap in knowledge (scientific research) should be clearly identified. It is necessary to define the research question more clearly and derive objectives from it (Introduction). Additionally, align the objectives in the Abstract with those in the Introduction for greater clarity.

References used in Literature review are mostly from recent publication and relevant.

Four research hypotheses have been established to be demonstrated during the modeling (382-395), but there is no explanation why are they important for the model and need to be tested (gap in knowledge).

Research results are reproducible based on the details given in the method section, and figures, tables are appropriate in a way to be easy understand.

It is recommended to provide a more in-depth discussion. This means explaining how your results contribute to the existing body of knowledge in your field. In this case, the hypotheses that are verified and those that are not must be included.

Future lines of research and implementation in other countries should be included in Conclusion. It is key to make a call for future related work that can expand your contribution.

Author Response

The Article titled 'The Modeling of Impact of VAT Fiscality on Branch-Level 2 Performance in the Construction Industry. Evidence 3 from Romania' is analysing the evolution of VAT fiscality in Romania and its impact on the performance in the construction sector. The main contribution and strength of the paper is in the proposed econometric model for assessing the impact of VAT on the business environment in Romania's construction sector based on determining annual models of the correlated evolution of deductible and collected VAT. The research concludes that possible changes in the VAT fiscal regime to achieve sustainability should have as little frequency as possible to ensure the necessary and sufficient predictability for the expected degree of voluntary compliance.

The research topic under investigation is current, the manuscript follows the proposed structure of the journal but some modifications are required. These recommendations are intended to bolster the scientific rigor, clarity, and overall quality of the document.

Dear reviewer,

Thank you for your relevant observations that have helped us improve the quality of the manuscript. We hereby present the changes made in accordance with your suggestions.

In the Abstract, it is meaningful to present the results of own research.

Authors: The abstract has been enlarged in accordance with the recommendations you provided. Additionally, further details and content have been incorporated to enhance its comprehensiveness.

The relevance of the manuscript should be explained by appropriate references to define main research questions especially in Introduction where no cited references (mostly recent publications within the last 5 years) were used (just one source). By briefly presenting the most important and relevant references of the review topic the gap in knowledge (scientific research) should be clearly identified. It is necessary to define the research question more clearly and derive objectives from it (Introduction). Additionally, align the objectives in the Abstract with those in the Introduction for greater clarity.

Authors: Additional clarifications regarding the relevance of the research have been introduced in the introduction, and the research questions have been formulated. Furthermore, the research objectives have been readjusted in accordance with the research questions and correlated with the objectives outlined in the abstract. The knowledge gap has been highlighted in the introduction.

 References used in Literature review are mostly from recent publication and relevant.

Four research hypotheses have been established to be demonstrated during the modeling (382-395), but there is no explanation why are they important for the model and need to be tested (gap in knowledge).

Authors: We have made the necessary completions in accordance with your suggestions.

Research results are reproducible based on the details given in the method section, and figures, tables are appropriate in a way to be easy understand.

It is recommended to provide a more in-depth discussion. This means explaining how your results contribute to the existing body of knowledge in your field. In this case, the hypotheses that are verified and those that are not must be included.

Authors: Following your suggestions, at the end of Chapter 4, we explain the contribution to the specialized literature made by the research conducted by the authors, exemplifying the observed effects.

Future lines of research and implementation in other countries should be included in Conclusion. It is key to make a call for future related work that can expand your contribution.

Authors: Have been included the limitations of the study and future research directions in accordance with your suggestions.

Reviewer 2 Report

Comments and Suggestions for Authors 

Major comments:

Methods

1.     The biggest concern relates to the research methodology that breaks the panel into individual years which makes it impossible to study the impact of the introduction of sector-specific policies such as the reverse charge mechanism that may have influenced annual fluctuations in VAT evasion. In fact there is a slew of confounding factors that should be accounted for based on the authors’ institutional background information  such as limitation of deductibility, changes in the VAT regime (2010), restriction of the right to deduct cars and fuels (2011), formation of the fiscal group (2012), VAT liability upon collection (2012), reduction of the tax base in case of bankruptcy (2013), transformation of the VAT cash accounting system into an optional system (2013), tax rate changes in 2017. Therefore, the paper should employ methods for panel data (e.g. year fixed effects and firm or tax administrative office fixed effects) to analyze the relationships of interest and directly account for these policy changes in the study timeframe.

2.     Use of large firms. The choice of the authors to consider only firms employing more than 100 individuals is skewing the sample of firms towards larger and potentially less susceptible to VAT evasion. In addition, the authors do not discuss how representative of the VAT performance in the Romanian construction industry, in its entirety, this study sample of relatively large firms is. The inclusion criteria of firms should be motivated better and the earlier comments should be discussed in detail.

Presentation of results

3.     In addition, the study needs to conform to the basic standards of presenting equations and results. The authors should refrain from including the actual estimated coefficients in the structured equations. These should not be reported outright in each listed equation. This will help reduce the page count substantially as there is unnecessary white space devoted to reporting coefficient estimates in raw form. Instead, the authors should only be providing with the average estimated effects in the regression output tables, more succinctly.

Minor comments:

1.     Literature Review. The literature review is reasonably up to date. There could be some improvements in the VAT performance section in response to the rise in the use of credit and debit cards in Madzharova (2020). There is a very relevant but currently unpublished work on the Effects of the Reverse Charge Mechanism on the VAT Gap focusing on the construction sector in Italy. It shows that a substantial VAT evasion was deterred after the RCM was introduced to housing sector transactions. This study would be interesting to elude to and at least mention parallel work that is worth bringing up as it provides with the opportunity of comparing and contrasting the Romanian case to the Italian estimates: Another work that considers VAT gaps and how the use of e-money and cash is Bohne, A., Koumpias, A., & Tassi, A. (2023).      

2.        Institutional background. The authors should give an idea to the reader of how Romania compares relative to other EU countries in terms of VAT evasion. There are estimates of the VAT gaps that are available for the study years and all EU member states that will be of relevance to the reader. It is mentioned that Romania has recorded relatively high economic growth “in the consumer economy” but it is not clear if increased VAT collections reflect reduced rates or increased VAT compliance.

3.     Presentation of Results. It also of paramount importance that the exposition of the model results to be revised. The authors are reporting histograms and correlation coefficient estimates that makes presentation very tedious. By merging year-to-year information to a single panel, the visual representation of results could be achieved with a single figure which is more tractable for the reader.

4.     Make sure you use the appropriate name is reverse charge mechanism or RCM, not just reverse charge when referring to the policy. 

References

Bohne, A., Koumpias, A., & Tassi, A. (2023). Cashless Payments and Tax Evasion: Evidence From VAT Gaps in the EU. ZEW-Centre for European Economic Research Discussion Paper, (60).

Madzharova, B. (2020). Traceable payments and VAT design: Effects on VAT performance. CESifo Economic Studies, 66(3), 221-247.

Stiller, W., & Heinemann, M. (2019). Do more harm than good? The optional reverse charge mechanism against VAT fraud. Preprint on Research Gate. https://doi. org/10.13140.

Comments on the Quality of English Language

Language should be substantially improved. Emphasis should be placed on improving the flow of the reading and drop unnecessary math equations.

Author Response

Dear reviewer,

We express our gratitude for your insightful observations, which have significantly contributed to enhancing the overall quality of the manuscript. In response to your valuable suggestions, we are pleased to present the following modifications implemented to align with your feedback.

Major comments:

Methods

  1. The biggest concern relates to the research methodology that breaks the panel into individual years which makes it impossible to study the impact of the introduction of sector-specific policies such as the reverse charge mechanism that may have influenced annual fluctuations in VAT evasion. In fact there is a slew of confounding factors that should be accounted for based on the authors’ institutional background information  such as limitation of deductibility, changes in the VAT regime (2010), restriction of the right to deduct cars and fuels (2011), formation of the fiscal group (2012), VAT liability upon collection (2012), reduction of the tax base in case of bankruptcy (2013), transformation of the VAT cash accounting system into an optional system (2013), tax rate changes in 2017. Therefore, the paper should employ methods for panel data (e.g. year fixed effects and firm or tax administrative office fixed effects) to analyze the relationships of interest and directly account for these policy changes in the study timeframe.

 Authors: We thank you for your valuable suggestions that have significantly contributed to the improvement of our research. We have modified the methodology, introduced the Panel approach, and incorporated the structural equations model in Figures 1 and 2. We conducted the Time-fixed effects calculation according to the data in Table 4. Analyzing the above-mentioned modifications suggested by you, a correlation is observed between changes in fiscal policy and the model's outcomes.

  1. Use of large firms. The choice of the authors to consider only firms employing more than 100 individuals is skewing the sample of firms towards larger and potentially less susceptible to VAT evasion. In addition, the authors do not discuss how representative of the VAT performance in the Romanian construction industry, in its entirety, this study sample of relatively large firms is. The inclusion criteria of firms should be motivated better, and the earlier comments should be discussed in detail.

Authors: We introduced limitations regarding the sample selection method in the conclusions, and we defined future research directions to expand the conclusions in accordance with your relevant observations.

Presentation of results

  1. In addition, the study needs to conform to the basic standards of presenting equations and results. The authors should refrain from including the actual estimated coefficients in the structured equations. These should not be reported outright in each listed equation. This will help reduce the page count substantially as there is unnecessary white space devoted to reporting coefficient estimates in raw form. Instead, the authors should only be providing with the average estimated effects in the regression output tables, more succinctly.

Authors: We have made all the changes according to your suggestions, for which we extend our respectful thanks.

 Minor comments:

  1. Literature Review. The literature review is reasonably up to date. There could be some improvements in the VAT performance section in response to the rise in the use of credit and debit cards in Madzharova (2020). There is a very relevant but currently unpublished work on the Effects of the Reverse Charge Mechanism on the VAT Gap focusing on the construction sector in Italy. It shows that a substantial VAT evasion was deterred after the RCM was introduced to housing sector transactions. This study would be interesting to elude to and at least mention parallel work that is worth bringing up as it provides with the opportunity of comparing and contrasting the Romanian case to the Italian estimates: Another work that considers VAT gaps and how the use of e-money and cash is Bohne, A., Koumpias, A., & Tassi, A. (2023).

Authors: Based on your suggestions, we have expanded the study of specialized literature and referred to the relevant sources suggested by you.

  1. Institutional background. The authors should give an idea to the reader of how Romania compares relative to other EU countries in terms of VAT evasion. There are estimates of the VAT gaps that are available for the study years and all EU member states that will be of relevance to the reader. It is mentioned that Romania has recorded relatively high economic growth “in the consumer economy” but it is not clear if increased VAT collections reflect reduced rates or increased VAT compliance.

Authors: Based on your suggestions, we have completed the introductory chapter with relevant information regarding the VAT gap in Europe, providing details about Romania's position in the European ranking compared to other member states.

  1. Presentation of Results. It also of paramount importance that the exposition of the model results to be revised. The authors are reporting histograms and correlation coefficient estimates that makes presentation very tedious. By merging year-to-year information to a single panel, the visual representation of results could be achieved with a single figure which is more tractable for the reader.

Authors: According to your suggestions, we have modified the approach, leading to more relevant results. Thank you.

  1. Make sure you use the appropriate name is reverse charge mechanism or RCM, not just reverse charge when referring to the policy.

Authors: We have checked the text and made the necessary modifications.

References

Bohne, A., Koumpias, A., & Tassi, A. (2023). Cashless Payments and Tax Evasion: Evidence From VAT Gaps in the EU. ZEW-Centre for European Economic Research Discussion Paper, (60).

Madzharova, B. (2020). Traceable payments and VAT design: Effects on VAT performance. CESifo Economic Studies, 66(3), 221-247.

Stiller, W., & Heinemann, M. (2019). Do more harm than good? The optional reverse charge mechanism against VAT fraud. Preprint on Research Gate. https://doi. org/10.13140.

  Authors: The references you indicated have been included in the text.

Comments on the Quality of English Language

Language should be substantially improved. Emphasis should be placed on improving the flow of the reading and drop unnecessary math equations.

 Authors We have made the necessary corrections. Several paragraphs have been rewritten to improve the flow of the reading. Unnecessary equations have been removed.

Round 2

Reviewer 2 Report

Comments and Suggestions for Authors

The revised paper has been significantly improved to its earlier. No further comments to address. 

Comments on the Quality of English Language

The authors should try to be more succinct in their arguments presented in the discussion section.